



# Sensitivity of Pliocene climate simulations in MRI-CGCM2.3 to respective boundary conditions

Youichi Kamae[1,2], Kohei Yoshida[3], Hiroaki Ueda[1]

[1]Faculty of Life and Environmental Sciences, University of Tsukuba, Tsukuba, 305-8572, Japan
[2]Scripps Institution of Oceanography, University of California San Diego, La Jolla, 92093-0206, USA
[3]Meteorological Research Institute, Tsukuba, 305-0052, Japan

*Correspondence to*: Youichi Kamae (kamae.yoichi.fw@u.tsukuba.ac.jp)

**Abstract.** Accumulations of global proxy data are essential steps for improving reliability of climate model simulations for the Pliocene warming climate. In the Pliocene Model Intercomprison Project phase 2 (PlioMIP2), boundary forcing data
have been updated from phase 1 due to recent advances in understanding of oceanic, terrestrial and cryospheric aspects of the Pliocene paleoenvironment. In this study, sensitivities of Pliocene climate simulations to the newly archived boundary conditions are evaluated by a set of simulations using an atmosphere-ocean coupled general circulation model, MRI-CGCM2.3. The simulated Pliocene climate is warmer than pre-industrial condition for 2.4 K in global mean, corresponding to 0.6 K warmer than the PlioMIP1 simulation by the identical climate model. Revised orography, lakes and shrunk ice
sheets compared with the PlioMIP1 lead to local and remote influences including snow and sea ice albedo feedback, and poleward heat transport due to the atmosphere and ocean that result in additional warming over middle and high latitudes. The amplified higher-latitude warming is supported qualitatively by the proxy evidences, but is still underestimated quantitatively. Physical processes responsible for the global and regional climate changes should be further addressed in future studies under systematic intermodel and data-model comparison frameworks.

**1 Introduction**

Atmosphere-ocean coupled general circulation models (AOGCMs) have widely been used for climate projections on decadal to centennial timescales since the late 20th century. Recently, representations of large-scale climate state, its variability, and parameterized processes in the AOGCMs (e.g. cloud and convection) have been improved substantially (e.g. Reichler and Kim, 2008; Klein et al., 2013; Bellenger et al., 2014). A large ensemble of multiple climate models has contributed to
addressing robust climate trends in the future projections (e.g. Xie et al., 2015). Model intercomparison of the past climate changes and data-model comparisons are powerful frameworks for understanding physical processes responsible for climate change and variability and assessing reliability of the climate model projections (e.g. Braconnot et al., 2012; Masson-Delmotte et al., 2014).





Since the 1990s, globally warmed climate during the Pliocene (~5 to 3 Ma) has attracted much attention as a potential analogue for the ongoing global climate change (e.g. Masson-Delmotte et al., 2014). Previous modelling and proxy-based studies revealed that the Pliocene climate can be characterized by substantial global warming (2.7 to 4.0 K) with anomalous zonal and meridional temperature gradients (Dowsett et al., 1992, 2010; Wara et al., 2005; Fedorov et al., 2013; Haywood et

al., 2013, 2016a). Under the Pliocene Model Intercomprison Project phase 1 (PlioMIP1; Haywood et al., 2010, 2011), a part project of the Paleoclimate Modelling Intercomparison Project phase 3 (PMIP3), nine AOGCMs provided results of mid-Pliocene (3.264 to 3.025 Ma) climate simulations. A paleoenvironmental reconstruction project named the Pliocene Research Interpretation and Synoptic Mapping (PRISM) for PlioMIP1 (PRISM3D; Dowsett et al., 2010) provided global boundary condition dataset that is needed for performing the mid-Pliocene climate simulations. Through intercomparison of

multiple model results and data-model comparisons, large-scale climate pattern (Haywood et al., 2013), East Asian monsoon behaviour (R. Zhang et al., 2013), Atlantic Meridional Overturning Circulation (AMOC; Z.-S. Zhang et al., 2013), terrestrial climate (Salzmann et al., 2013) and oceanic conditions (Dowsett et al., 2012, 2013) were examined systematically. The PlioMIP1 was the first model intercomparison project comparing past modelled climate prescribed with proxy-based vegetation pattern that is distinct from the present-day condition. Numerous independent proxy data for the Pliocene

(Haywood et al., 2016a) suggested predominant warming over the North Atlantic and the Arctic region and wetter climate over land (Salzmann et al., 2008). The climate simulations under the PlioMIP1 tended to underestimate the warming over the Northern Hemisphere middle and high latitudes (Salzmann et al., 2013; Dowsett et al., 2012, 2013) although the latitudinal warming gradient was reproduced qualitatively. Hill et al. (2014) pointed out a robust contribution of surface albedo due to changes in vegetation and ice sheets and ice-albedo feedbacks (sea ice and snow) to the warming amplifications over the

polar regions.

Since the PRISM3D/PlioMIP1, newly archived proxy evidences have been integrated as PRISM4 dataset (Dowsett et al., 2016; hereafter D16) that is planned to be used for an ongoing modelling intercomparison project, PlioMIP phase 2 (PlioMIP2; Haywood et al., 2016b; hereafter H16b). In the PRISM4/PlioMIP2, boundary conditions for the Pliocene climate simulations including orography and ice sheets have been updated. In addition, global lakes and soil data were newly

included in the PRISM4 dataset. Numerous mega-lakes over land associated with the wetter terrestrial climate during the Pliocene were suggested to be important for simulating local and large-scale anomalous climate (Pound et al., 2014).

In the PlioMIP2, dynamical predictions of vegetation and lakes, changes in land and ocean topography, and change in the land-sea mask are recommended for the Pliocene climate simulations. However, the respective roles of the revised boundary conditions in the Pliocene climate simulations have not been addressed sufficiently. A set of sensitivity experiments with

different combinations of modern and Pliocene boundary conditions is planned to be performed in the PlioMIP2. The respective roles of the boundary conditions can be evaluated by comparing results of the sensitivity runs. In this study, we conduct PlioMIP2 climate simulations by using an AOGCM, MRI-CGCM2.3, that was also used in the PlioMIP1 (Kamae and Ueda, 2012; hereafter KU12). The results of the PlioMIP1 run, the PlioMIP2 run, and the PlioMIP2 sensitivity runs with





swapped boundary conditions are used to examine respective roles of the updated boundary conditions. This study reports that the revised ice sheets in the high latitudes and global land properties (orography and lakes) result in an amplified high latitude warming via direct influence, radiative feedback, and atmospheric and oceanic heat transports. The anomalous warming simulated in the PlioMIP2 protocol is more consistent with the proxy evidences than the PlioMIP1 run. Section 2

describes the data and methods including proxy-based boundary conditions and modelling strategy. Section 3 presents general characteristics of simulated Pliocene climate and compares it with the PlioMIP1 results. Section 4 examines the roles of atmospheric and oceanic meridional heat transports in the simulated middle and high latitude warming in the model. Section 5 compares modelled and reconstructed SST and assess SST reproducibility of the model simulation. In Sect. 6, we present a summary and discussion of this study.

**2 Data and Methods**

**2.1 Climate model**

An AOGCM named MRI-CGCM2.3 (Yukimoto et al., 2006) was used for performing the PlioMIP2 experiments in this study. The model is identical to that used in the PlioMIP1 (KU12). Atmospheric model has a horizontally T42 resolution (~2.8°) and vertically 30 layers (model top is 0.4 hPa). Oceanic component is a Bryan-Cox-type ocean general circulation

model with a horizontal resolution of 2.5° longitude and 2.0°–0.5° latitude and 23 layers (the deepest layer is 5000 m). Details of the atmosphere and ocean models can be found in Yukimoto et al. (2006) and KU12. Vegetation, lakes and atmospheric $CO_2$ concentration are prescribed in the model as boundary conditions (see Sect. 2.2) because the model does not predict vegetation, lakes, and carbon cycle (Table 1). Land scheme is simple biosphere model (SiB2; Sellers et al., 1986; Sato et al., 1989), which predicts soil water and surface heat budget. Parameters for the land scheme depend on 13 types of

vegetation category (see Sect. 2.2). The model predicts water budget for lakes, but the lake surface temperature is predicted by the heat budget at the water surface, assuming a slab with a thickness of 50 m (Yukimoto et al., 2006).

Although both the AOGCM and an atmosphere-only general circulation model were used in the PlioMIP1 (Kamae et al., 2011; KU12), we perform only the AOGCM simulations in this study according to the PlioMIP2 protocol (H16b). In KU12, a set of AOGCM simulations with and without flux adjustments (heat, fresh water flux and wind stress) were performed. In

the present study, we only integrated the model without any flux adjustments (similar to AOGCM_NFA run in KU12).

**2.2 Experimental designs for pre-industrial and Pliocene climate simulations**

In addition to two core experiments (pre-industrial and Pliocene), a set of sensitivity experiments (Table 2) is also proposed in the PlioMIP2 (Table 3 in H16b). Alterations of soil and land-sea mask (e.g. the Bering Strait and the Canadian Arctic Archipelago) were recommended in the PlioMIP2. Due to technical difficulties, these alterations were not incorporated into

the current PlioMIP2 simulations. Due to the low resolution and the simple experimental setting (i.e. low computational cost), we can conduct all the set of simulations proposed in the PlioMIP2 (totally 12-type 500-yr long simulations; H16b). We also



plan to conduct higher resolution and/or higher complexity PlioMIP2 simulations with a higher-resolution AOGCM and/or an Earth system model (see Sect. 6).

We can compare results of this study with the PlioMIP1 directly because the experimental setting for the PlioMIP2 pre-industrial run (Tables 1 and 2) is identical to the PlioMIP1 (KU12). In the pre-industrial run, $CO_2$, $CH_4$ and $N_2O$

concentrations were set to be 280 ppmv, 760 ppbv and 270 ppbv, respectively. Ozone concentration in each month was derived from climatology in Wang et al. (1995). Orbital parameters were identical to the PlioMIP1 (Table 1). Modern land orography, lakes, and ice sheet of the MRI-CGCM2.3 (KU12; Figs. 1a, 2a) were used in the pre-industrial run. Vegetation pattern was derived from PRISM3D modern map converted into 13 types of SiB2 classification (Table 3 in KU12; Fig. 1a).

To simulate the Pliocene climate, the PRISM4 global paleoenvironmental dataset (D16) including atmospheric trace gases,

orography, vegetation covers, ice sheets, and lakes are prescribed in the model. Pliocene orography was updated from the PRISM3D (Sohl et al., 2009) by considering mantle flow (Rowley et al., 2013) and glacial isostatic response of ice sheet loading (Raymo et al., 2011). We added anomalous orography (Pliocene minus modern) to the model's modern orography (Fig. 3) according to the "anomaly method" recommended in the PlioMIP2 (H16b). Direct proxy evidences for the Greenland and Antarctic ice sheets during the Pliocene were not available. The PRISM4 provided Greenland ice sheet data

that is confined to high elevations in the East Greenland Mountains suggested by results of Pliocene Land Ice Sheet Model Intercomparison Project (Dolan et al., 2015; Koenig et al., 2015). According to proxy data for the Antarctic ice sheet (Naish et al., 2009; Pollard and DeConto, 2009), ice-free condition and the identical ice sheet estimate to the PRISM3D were assumed in the West and East Antarctica, respectively (D16; Figs. 1b, 3). The resultant prescribed ice sheets over the Greenland and Antarctica are smaller than that in KU12. Because the model does not predict dynamic vegetation, we

prescribe vegetation pattern (Fig. 1b) that is identical to the PRISM3D/PlioMIP1 (Salzmann et al., 2008), according to the recommendation in H16b.

In addition to vegetation, global lake distribution during the Pliocene was also suggested to be distinct from the present day. Schuster et al. (2001, 2006) and Griffin (2006) suggested the existence of the African Megalakes existed during the Miocene and the Pliocene. Contoux et al. (2013) and Pound et al. (2014) pointed out an important role of the Pliocene lakes in the

global climate through local atmosphere-land interaction and remote influences. The PRISM4 provides global lake area data (Pound et al., 2014) for the PlioMIP2 (Fig. 2). We prescribe the Pliocene lakes (Fig. 2b; Table 2) by adding anomalous areas of lakes (Fig. 2c) to model's modern lakes (Fig. 2a).

In this paper, the results of six PlioMIP2 experiments (Table 2) are reported: pre-industrial run; Pliocene run; pre-industrial runs but with $CO_2$ concentration of 400 and 560 ppmv (hereafter E400 and E560 runs); pre-industrial run but with Pliocene

ice sheet (Ei280 run); and pre-industrial run but with Pliocene orography, lakes, and vegetation (OVL; hereafter Eo280 run). $CH_4$, $N_2O$, ozone, solar constant, orbital parameters are identical to the pre-industrial run (Table 1). Land orography, lake area, vegetation, land ice, atmospheric $CO_2$ concentration are altered in the Pliocene run from the pre-industrial run.



Combinations of the boundary conditions are changed in the sensitivity runs so that the impacts of the individual components can be evaluated (Table 2; Sect. 2.3). A large spread remains in the assessment of the Pliocene $CO_2$ concentration (e.g. Raymo et al., 1996; Seki et al., 2010; D16). Although 400 ppmv of $CO_2$ concentration is prescribed in the Pliocene run, other concentrations are also used in the other sensitivity runs (H16b). In this paper, we examine the results of simulations with

$CO_2$ concentration of 280, 400 and 560 ppmv (Table 2).

The results are also compared with PlioMIP1 run conducted in KU12 (Table 2). Pliocene lakes were set to be identical to the modern. The PRISM3D-based vegetation (Salzmann et al., 2008) was also prescribed in the PlioMIP1 Pliocene run (Fig. 1b). A prescribed $CO_2$ concentration of 405 ppmv was slightly higher than the PlioMIP2.

The initial condition for the PlioMIP2 runs is identical to the pre-industrial run: 31 December of the PlioMIP1 control run (in

NFA_AOGCM) after 500-yr spin up (Fig. 3b in KU12). The model is integrated for another 500 years with swapped boundary conditions and the last 50 years are used for analyses. Note that reconstructed deep ocean temperature was added to the initial condition of the PlioMIP1 Pliocene run (KU12), distinct to the PlioMIP2.

### 2.3 Respective roles of boundary conditions

In the PlioMIP2, relative contributions of the boundary conditions to climate anomalies (e.g. global mean warming) can be

evaluated by comparing the set of sensitivity experiments (H16b). In this study, we use the six PlioMIP2 simulations (Table 2) and evaluate the respective contributions by following Eqs. (1–6):

$$All = [Eoi400] - [E280] \, , \tag{1}$$

$$CO2 = [E400] - [E280] \, , \tag{2}$$

$$OVL = [Eo280] - [E280] \, , \tag{3}$$

$$Ice\ Sheet = [Ei280] - [E280] \, , \tag{4}$$

$$Sum = CO2 + OVL + Ice\ Sheet \, , \tag{5}$$

$$Residual = All - Sum \, , \tag{6}$$

where [] represents the experiment name (Table 2). E280 and Eoi400 indicate the pre-industrial and Pliocene runs, respectively. Difference of results between Eo280 and E280 run corresponds to effect of differences in OVL. Sum of Eqs.

(2–4) is used as a reconstruction of Eq. (1). By Eqs. (5, 6), we can compare the simulated climate anomaly between the Pliocene run and the pre-industrial and its reconstruction. *Residual* (Eq. 6) indicates a nonlinear effect of the boundary conditions. This decomposition method is not identical to that recommended in H16b. While decomposed relative contributions could be dependent on the choice of quantifying methods, *Residual* term shown in this study is generally minor to *All* (see Sect. 3), suggesting an effectiveness of the decomposing method used in this study.





## 3 Results

### 3.1 Global mean warming in Pliocene run

First, we compare global mean surface air temperature (SAT) change between the simulations. Figure 4 shows the time evolution of the global mean SAT during the 500-yr model integrations. Compared with the pre-industrial run, identical to
the PlioMIP1 control simulation (KU12), all the experiments show higher global mean SAT. The Pliocene run shows a more stable long-term trend (year 70–500) than other sensitivity runs prescribed with the OVL (Eo280) or $CO_2$ (E400). The doubling $CO_2$ experiment (E560) shows a long-term warming trend and the resultant warming for the last 50 years accounts for 2.8 K (Table 3). In the E400 run prescribed with $CO_2$ concentration of 400 ppmv, global mean warming accounts for 1.7 K that is the largest contributor to the Pliocene warmth among the boundary conditions (68 %; Table 3). Here *Sum* (2.5 K)
can reconstruct *All* (2.4 K) quantitatively. Contributions of *Ice Sheet* and *OVL* are 12 and 20 %, respectively. Compared with the PlioMIP1 run (1.8 K), the PlioMIP2 Pliocene run shows a larger warming (+39 %, 0.7 K) although the prescribed $CO_2$ concentration (400 ppmv) is slightly lower (405 ppmv in PlioMIP1). In the next section, we compare spatial patterns of the results and decompose respective contributions of the individual boundary conditions.

### 3.2 Regional changes in temperature and precipitation

Figure 5 shows spatial distributions of annual mean SAT anomalies averaged over the last 50 years. Similar to the PlioMIP1 run (KU12; Haywood et al., 2013), the Pliocene anomaly exhibits a polar amplification of surface warming (i.e. warming peaks over the high latitudes including Greenland and the Antarctica). While land surface warming is generally larger than the ocean surface warming (in response to $CO_2$ forcing; Fig. 5b; Manabe et al., 1991; Kamae et al., 2014), regional differences in SAT change (weak warming compared with surrounding areas) are found over southern North America,
tropical Africa, Indian subcontinent, and eastern Siberia, similar to the PlioMIP1 (KU12). Zonally averaged SAT change exhibits (1) minimum warming over the Southern Hemisphere middle latitude; (2) moderate warming over the tropics; and (3) warming peaks over the Southern and Northern Hemisphere high latitudes (Fig. 6a). Sea surface temperature (SST) also exhibits the inter-hemispheric warming asymmetry (Figs. 5e, 6b; see Sect. 3.3) and warming peak in the Northern Hemisphere mid-to-high-latitude (particularly in the eastern North Pacific and North Atlantic; Figs. 5e, 6b). Here *Sum* of
zonal mean SAT, SST and other variables (e.g. precipitation) are similar to *All* (Fig. 6), indicating a limited *Residual* term in the zonal mean (Eq. 6). The effectiveness of reconstruction of *All* by *Sum* suggests that characteristics of the Pliocene climate anomaly can be decomposed into the individual contributions by Eq. (5). Note that *Sum* tends to underestimate (overestimate) the Southern (Northern) Hemisphere middle and high latitude warming, suggesting an importance of nonlinear effects (see Sect. 4).

Compared to the PlioMIP1, the Pliocene run exhibits a larger warming over Antarctica and the Northern Hemisphere middle and high latitudes (Fig. 6a), resulting in the larger increase in global mean SAT (Sect. 3.1). In addition to the less ice sheets over Greenland and West Antarctica, other factors also contribute to the polar amplification of surface warming (Figs. 5, 6a).





The decomposition based on the sensitivity runs (Sect. 2.3) indicates that all the boundary conditions contribute to the polar warming (Figs. 5, 6a). Contributions of *OVL* and *CO2* to the zonal mean polar warming are dominant (Figs. 5b, c, 6a). Here *OVL* is the largest contributor to the latitudinal difference in the Northern Hemisphere warming and the inter-hemispheric warming contrast (Fig. 6a). Although spatially smooth $CO_2$ radiative forcing also leads to the polar amplification (Fig. 6a; e.g. Serreze and Barry, 2011; Hill et al., 2014), *OVL* effect dominates the meridional warming contrast. Note that *CO2* is the largest contributor to the global mean Pliocene warming (Sect. 3.1). In Sect. 5, we assess reproducibility of the SST in the PlioMIP2 and 1 runs by comparing with proxy-based estimate.

Figures 7 and 6e show change in surface albedo and its zonal mean. Increasing albedo over North America middle latitude and eastern Siberia due to change in vegetation (boreal forest in modern but grassland in the Pliocene; Fig. 1; Haywood et al., 2013; Hill et al., 2014), a part of *OVL* effect, contributes to the regional difference in the SAT change (Figs. 5c, 7c). Over the high latitude (deciduous conifer, tundra and bare soil regions over northern Canada and northeastern Eurasia; Fig. 1), northward shift of boreal forest leads to a reduction of surface albedo (Figs. 6e, 7c) and resultant regional warming (Fig. 5c). Surface snow cover (not shown) also affects partly the land surface albedo. Decreasing albedo over the high latitude ocean (the Arctic and Antarctic Ocean) corresponds to sea ice reductions (Fig. 6e, f) that are larger than the PlioMIP1 (Fig. 6f; Howell et al., 2016). Sea ice and snow albedo feedback contributes to the differential polar and global-mean warming between the two runs. Over semi-arid and arid land regions, local SAT change corresponds well with precipitation change (Fig. 8; e.g. Kamae et al., 2011). *OVL* leads to increased precipitation over western North America, tropical Africa, and Indian subcontinent (Fig. 8c), resulting in surface cooling (Fig. 5c) via changing surface heat fluxes (not shown).

The precipitation response in the PlioMIP2 run, dominated by *OVL* effect, is generally larger than the PlioMIP1 run (Figs. 6c, 8; Fig. 7g in KU12). The anomalous middle and high latitude warming could affect the large-scale precipitation pattern via changing atmospheric circulations (Sect. 3.3). For example, Atlantic interhemispheric warming gradient (warmer in the North Atlantic than the South Atlantic; Fig. 5g) affects the tropical Atlantic precipitation (Fig. 8c; e.g. Zhang and Delworth, 2006). In addition, regional alterations of land surface condition can also affect local precipitation. The expansions of the lakes over tropical Africa and mid-latitude western North America (Fig. 2c) reduce surface sensible heat flux and enhance local hydrological cycle (e.g. surface evaporation and precipitation; Pound et al., 2014). The enhanced precipitation over the semi-arid regions is an important factor for simulating the Pliocene vegetation pattern (Kamae and Ueda, 2011; Contoux et al., 2013; Pound et al., 2014).

### 3.3 Meridional overturning circulation

The surface temperature (SAT and SST) shows warming peaks over the middle and high latitudes and the interhemispheric warming asymmetry (Figs. 5, 6a, b). The warming peaks in the PlioMIP2 are larger than that the PlioMIP1 (Fig. 6a). Both changes in precipitation (Figs. 6c, 8) and cloud amount (not shown) show interhemispheric asymmetries (larger increase in the Northern Hemisphere than the Southern Hemisphere), suggesting a change in large-scale atmospheric circulation (e.g.



Kang et al., 2009). Figure 9 shows changes in atmospheric mean meridional circulation (MMC) determined by mass stream function (MSF). The MMC, one of the important factors for the Pliocene climate anomaly (Chandler et al., 1994; Brierley et al., 2009; Brierley and Fedorov, 2010; Kamae et al., 2011; Sun et al., 2013; Li et al., 2015), shows a larger change compared with the PlioMIP1 (Fig. 6d). The MMC change is largely characterized as enhanced (weakened) Southern (Northern) Hemisphere Hadley cell, northward shift of the tropical Hadley cells, and enhanced mid-latitude cell over the Northern Hemisphere (Figs. 6d, 9a). The boundary of the two Hadley cells and northern edge of the northern cell (determined by signs of MSF at 500 hPa level; Fig. 6d) shift northward for 7.2° and 2.0°, respectively, and the mid-latitude cell is enhanced for 41 %, largely according to the *OVL* effect (Figs. 6d, 9c). The change in the Hadley circulation is consistent with the PlioMIP1 qualitatively but larger quantitatively, suggesting an anomalous meridional heat transport due to the MMC (see Sect. 4).

The simulated SST anomaly shows the remarkable meridional warming gradient, particularly over the Atlantic (Fig. 5), suggesting a substantial anomaly in the AMOC. Climatological AMOC in the pre-industrial run shown in Fig. 10 (4.2 Sv) is weaker than observations (e.g. Buckley and Marshall, 2016) and other PlioMIP1 models (Z.-S. Zhang et al., 2013). Note that AMOC simulated in MRI-CGCM2.3 shown in Z.-S. Zhang et al. (2013) is a result of simulation with the flux adjustments (KU12). The Pliocene AMOC simulated in the PlioMIP2 (without any flux adjustments) is quite stronger (+15 Sv) than the pre-industrial run, suggesting an intensified northward heat transport due to the Atlantic. The AMOC change dominates in *OVL* (+14.5 Sv; Fig. 10c) and enhancement due to *CO2* is moderate. In the next section, we discuss possible factors contributing to the substantial higher latitude warming found in the Pliocene run.

**4 Mid- and high-latitude warming and meridional heat transport**

The PlioMIP2 run shows the larger middle and high latitude warming over the Northern Hemisphere (9 K at 75° N; Fig. 6a) compared with the PlioMIP1 run. Hill et al. (2014) evaluated the contributions of factors for the polar amplification by using eight PlioMIP1 models. Despite substantial inter-model spreads, strong warming due to reduced surface albedo was robustly found in all the models and relative contribution of meridional heat transport (due to the atmosphere and ocean) was minor. Figure 6g shows anomalous northward heat transport due to the atmosphere. Over the Southern Hemisphere high latitude and tropics to Northern Hemisphere middle latitude, anomalous southward heat transport can be found while the atmospheric heat transport is positive (northward) over the Southern Hemisphere middle latitude (30° S–55° S). The tropical southward heat transport is largely consistent with the MMC change (i.e. the intensified and weakened Southern and Northern Hemisphere Hadley cells; Figs. 6d, 9). *OVL* effect dominates and *CO2* effect is minor to the total change in the atmospheric heat transport. In contrast, *Ice Sheet* and *OVL* enhance northward heat transport over 50° N–70° N, contributing to the Arctic warming (70° N–80° N). Note that nonlinear *Residual* term is remarkable in the northward heat transport. The northward heat transport is quite limited in the PlioMIP1 run, consistent with the difference in the high latitude warming between the





PlioMIP2 and 1 (PlioMIP2 run shows stronger warming than PlioMIP1; Fig. 6a). The southward heat transport over the Southern Hemisphere high latitude contributes to the Antarctic amplification in the Pliocene run (6–7 K over 70° S–90° S). The simulated AMOC is much stronger than the pre-industrial run (Sect. 3.3), suggesting a substantial role in the North Atlantic warming during the Pliocene. Figure 6h shows northward heat transport due to the Atlantic Ocean. In contrast to

divergent responses among the PlioMIP1 models (Z.-S. Zhang et al., 2013), northward heat transport is enhanced substantially over the Northern Hemisphere (EQ–60° N). The enhanced heat transport is dominated by the *OVL* effect, consistent with the stronger AMOC found in the *OVL* (Fig. 10c). The stronger AMOC is also consistent with the substantial meridional SST gradient over the Atlantic (Fig. 5e), contributed by *OVL* and *CO2* (Fig. 5g, f). The enhanced mid-to-high-latitude warming is supported qualitatively by proxy-based SST reconstruction (Sect. 5).

The anomalous mid-to-high-latitude warming in the PlioMIP2 run is forced by the altered boundary conditions (Sect. 2.2) and is amplified/dampened by climate feedbacks and anomalous heat transports. As shown above, the impacts of $CO_2$ forcing and the Greenland ice sheet are limited (Figs. 5d, 6a, b), suggesting the importance of *OVL* in the mid-to-high-latitude warming including the North Atlantic and the eastern North Pacific (Fig. 5g). In addition to the enhanced northward heat transport due to the AMOC (Figs. 6h, 10c), *OVL* effect induces the anomalous atmospheric heat transport (Fig. 6g) via

changing atmospheric circulations (e.g. the Hadley circulation and eddy transport). The decreased surface albedo due to the northward shift of boreal forest in the Arctic region (60°N–80°N; Figs. 1, 6e, 7c) is an important external forcing due to *OVL*. In addition to the imposed albedo change, snow and sea ice albedo feedback also contributes to the predominant polar amplification (Figs. 5, 6a, b, e, f, 7). Note that *Sum* overestimates the sea ice reduction and surface warming simulated in *All*, indicating an importance of the nonlinear *Residual* term (Fig. 6a, b, f). Further analyses of relative contributions and inter-

model consistency of cloud and surface albedo, longwave radiation, meridional heat transport (Hill et al., 2014) due to atmosphere and individual ocean basins by using PlioMIP2 multi models may contribute to improve understanding of the physical mechanisms responsible for the Pliocene polar amplification.

## 5 Data-model comparison of SST

The Pliocene AMOC is apparently distinct from the pre-industrial run (Fig. 10), resulting in the anomalous northward heat

transport due to the Atlantic Ocean (Fig. 6h). Here the larger North Atlantic warming (3–7 K in 30° N–70° N; Fig. 5e) in the PlioMIP2 Pliocene run than the PlioMIP1 implies a better reproducibility of the mid-to-high-latitude SST warming that was robustly underestimated among the PlioMIP1 multi models (Dowsett et al., 2013). Figure 11 shows comparison of simulated SST and PRISM3D proxy-based SST reconstruction during the Pliocene (Dowsett et al., 2009). Note that the PRISM4 SST reconstruction is not updated (Dowsett et al., 2016) since PRISM3D. The SST reconstruction is characterized as extremely

high SST in the North Atlantic high latitude, low-to-mid-latitude warming gradient (limited change in the tropics and warming in the middle latitude, respectively), and remarkable warming in mid-latitude coastal areas (off the west coast of North America and South America, and off the east coast of the Eurasian Continent; Dowsett et al., 2009, 2013).



Figures 11 and 12 compare SST biases between the PlioMIP2 and 1. Generally, both the PlioMIP2 and 1 tend to underestimate the mid-to-high-latitude warming (Fig. 11a; Haywood et al., 2013). However, the large part of underestimation of the mid-to-high-latitude warming is reduced substantially (Fig. 11b). In contrast to the remarkable underestimation (SST bias is larger than 4 K; Fig. 12) of the mid-latitude warming (the North Atlantic, off the west coast of North America and South America, and off the east coast of the Eurasian Continent) in the PlioMIP1 (black circles in Figs. 11 and 12), the SST biases are much reduced in the PlioMIP2 run (Figs. 11b and 12). Note that SST bias (8.9–12 K in 69° N–81° N) over the North Atlantic high latitude (open blue circles in Figs. 11 and 12; Dowsett et al., 2013) is still not reduced in this simulation. From a zonal-mean perspective, the larger mid-to-high-latitude warming (Fig. 6a, b) in the PlioMIP2 run is more consistent with the proxy evidences than the PlioMIP1 (Figs. 11b, 12).

## 6 Summary and discussion

The PlioMIP2 simulations are conducted by using MRI-CGCM2.3 and prescribing the updated Pliocene paleoenvironmental dataset, called PRISM4. The Pliocene climate simulation with the identical model but with slightly revised boundary conditions from the PlioMIP1 results in the remarkable global-mean warming with the anomalous mid-to-high-latitude warming. The sensitivity experiments with swapped boundary conditions can largely reconstruct the modelled Pliocene climate anomalies, suggesting the linear additivity of the Pliocene climate simulation. The anomalous Northern Hemisphere higher-latitude warming can be understood as sum of direct response to the external forcing and associated climate feedbacks. The prescribed external forcing including $CO_2$, reduced ice sheet, and shortwave absorption due to the Arctic boreal forest contribute substantially to the higher-latitude warming. In addition, the anomalous northward heat transport associated with the large-scale atmospheric circulations and intensified AMOC, and snow and sea ice albedo feedback are also essential factors. The resultant anomalous warming over the mid-latitude ocean is more consistent with the proxy data than the PlioMIP1 simulation. However, the extremely warm condition over the Arctic to high-latitude North Atlantic region is not reproduced in this model.

The relative contributions to the polar amplification diverged substantially among multi models except those of surface albedo and $CO_2$ (Hill et al., 2014). In the PlioMIP1, the respective roles of the atmospheric and oceanic heat transports in the latitudinal warming gradient were not evaluated sufficiently. The intensified southern Hadley cell and a northward shift of the tropical cells can be confirmed in most of the PlioMIP1 models (Sun et al., 2013; Li et al., 2015). Both the PlioMIP1 multi models and the MRI-CGCM2.3 PlioMIP2 run show the predominant latitudinal contrast of surface warming over the Northern Hemisphere and the meridionally asymmetric change in the Hadley cells. However, the anomalous Hadley circulation transports heat southward, indicating that the change in the Hadley circulation is not a factor but can be understood as a result of the change in atmospheric meridional warming gradient (Li et al., 2015). Further analyses of the surface processes over land and ocean and three-dimensional atmospheric and oceanic processes (e.g. oceanic heat transport, atmospheric heat transport due to mean circulations, stationary eddies, transient eddies, feedback associated with ice albedo,



water vapour, cloud, and lapse rate; e.g. Serreze and Barry, 2011; Pithan and Mauritsen, 2014; Yoshimori et al., 2014) are needed to evaluate the respective contributions to the polar amplified climate in the Pliocene.

The simulated enhancement of the AMOC contributes to the North Atlantic warming in the Pliocene run. Z.-S. Zhang et al. (2013) revealed that none of the PlioMIP1 models simulated the substantial enhancement of the AMOC implied by the proxy

records (e.g. Raymo et al., 1996; Robinson, 2009). This is inconsistent with the current study because Z.-S. Zhang et al. (2013) introduced the results of flux-adjusted version of the MRI-CGCM2.3 model run (KU12). The MRI-CGCM2.3 without any flux adjustments simulates the much enhanced AMOC both in the PlioMIP1 and 2 settings (Fig. 6h). The current study points out the importance of *OVL* effect to the enhanced AMOC, but does not identify physical processes contributing to the drastic change. We plan to clarify the physical mechanisms by comparing spatial and vertical distribution

of salinity, heat and fresh water budget at sea surface, and its role in the AMOC strengths in the Pliocene run. Results of such analyses will be presented in a separated paper. Recent studies suggested that oceanic gateways (e.g. the Bering Strait; Hu et al., 2015) and bathymetry potentially contribute to past warming/cooling climate anomalies (e.g. Motoi et al., 2005; Robinson et al., 2011; Brierley and Fedorov, 2016). Sensitivity of simulated Pliocene climate to the PRISM4-based reconstructions of land-sea mask and bathymetry (D16) should be further evaluated in multi-model frameworks.

Assessment of regional climate properties (e.g. the Asian monsoon; R. Zhang et al., 2013) in the PlioMIP2 results is one of the remaining issues. Detailed data-model comparison of the oceanic and terrestrial climate should also be conducted to evaluate systematic biases in the PlioMIP2 model ensemble. The model experiment presented in the current study did not implement changes in the land-sea mask, soil, dynamic vegetation, and dynamic lakes. Implementation of the proxy-based soil properties as a boundary condition potentially affects the simulated Pliocene climate via changing surface and

atmospheric energy and water budget (Pound et al., 2014). Low resolution models are not suitable for simulating the regional atmospheric circulation and hydrological cycle associated with land orography (e.g. Xie et al., 2006). We plan to conduct more complex PlioMIP2 simulations that are more consistent with the proxy-based reconstructions by incorporating all the requested boundary conditions to an Earth system model or a high resolution AOGCM. Further complex and fine resolution modelling, multiple model intercomparison, and data-model comparisons could advance understanding of the factors for the

Pliocene warming climate.

**Acknowledgements**

The authors acknowledge PRISM4 and PRISM3D project members for archiving and providing global paleoenvironmental datasets for the Pliocene climate model simulations. We also thank A. M. Haywood and A. M. Dolan for coordinating the model intercomparison project, PlioMIP2.



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



**Table 1: Summary of model and experimental setting.**

| | |
|---|---|
| Model | MRI-CGCM2.3 |
| Paleogeography | Standard |
| Dynamic vegetation | No |
| Carbon cycle | No |
| Dynamical lake | No |
| $CH_4$ | 760 ppbv |
| $N_2O$ | 270 ppbv |
| Ozone | Wang et al. (1995) |
| Solar constant | 1365 W m$^{-2}$ |
| Eccentricity | 0.016724 |
| Obliquity | 23.446° |
| Perihelion | 102.04° |
| Integration length | 500 years |

**Table 2: Details of six PlioMIP2 experiments examined in this study. PlioMIP1 Pliocene run represents AOGCM_NFA run in Kamae and Ueda (2012). Soil and land-sea mask are identical among all the experiments.**

| Experiments | Orography, lakes | Vegetation | Ice sheet | CO$_2$ (ppmv) |
|---|---|---|---|---|
| E280 (Pre-industrial) | Modern | Modern | Modern | 280 |
| E400 | Modern | Modern | Modern | 400 |
| E560 | Modern | Modern | Modern | 560 |
| Ei280 | Modern | Modern | Pliocene | 280 |
| Eo280 | Pliocene | Pliocene | Modern | 280 |
| Eoi400 (Pliocene) | Pliocene | Pliocene | Pliocene | 400 |
| PlioMIP1 Pliocene | PlioMIP1 orography, modern lake | Pliocene | PlioMIP1 | 405 |



**Table 3: Anomalies in global-mean surface air temperature ($\Delta SAT$; K) in the PlioMIP2 runs relative to E280. The anomalies are calculated by averages for the last 50-yr of the individual runs.**

| Experiments | Global-mean $\Delta SAT$ (K) |
|---|---|
| E400 | 1.7 |
| E560 | 2.8 |
| Ei280 | 0.3 |
| Eo280 | 0.5 |
| Eoi400 | 2.4 |
| PlioMIP1 | 1.8 |





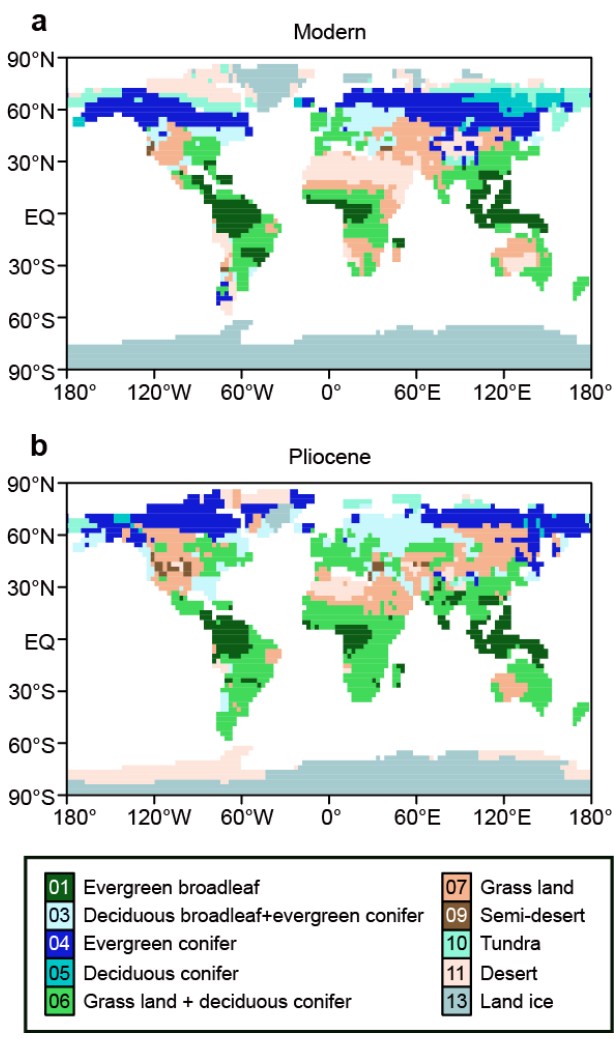

Figure 1: Prescribed land cover (SiB2 classification) for (a) modern and (b) Pliocene conditions.





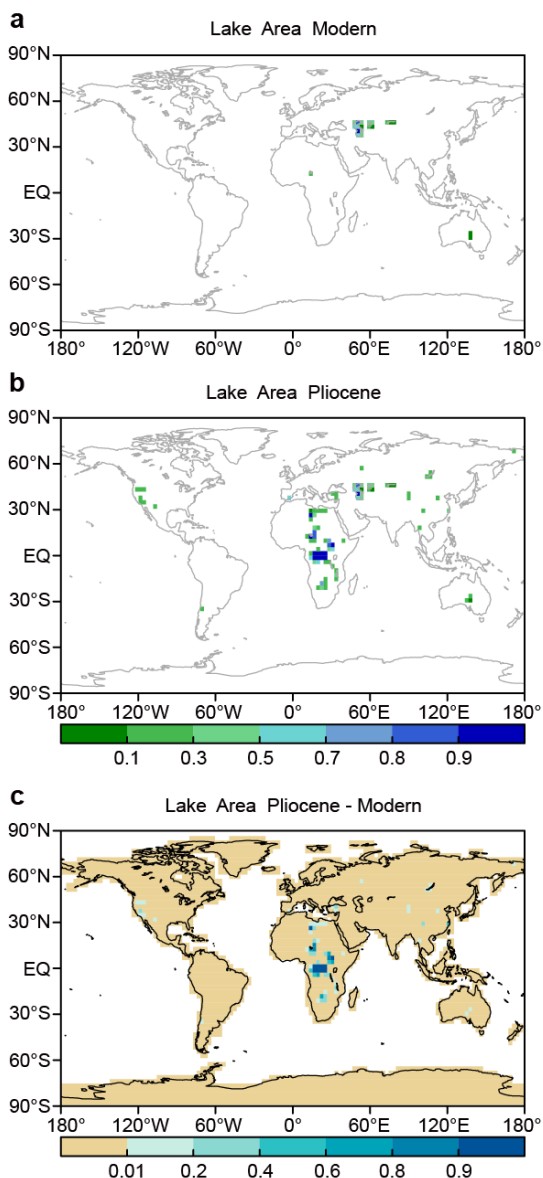

**Figure 2: Prescribed lake area fraction over land. (a) Modern, (b) Pliocene, and (c) Pliocene minus modern.**



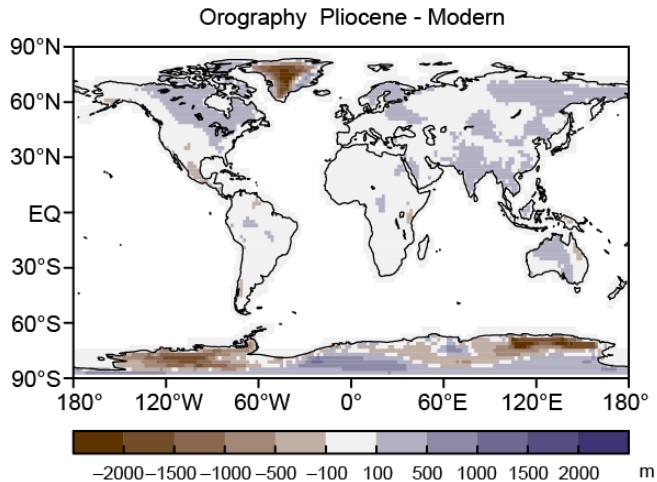

**Figure 3: Prescribed anomaly in land orography (m) for Pliocene relative to modern.**

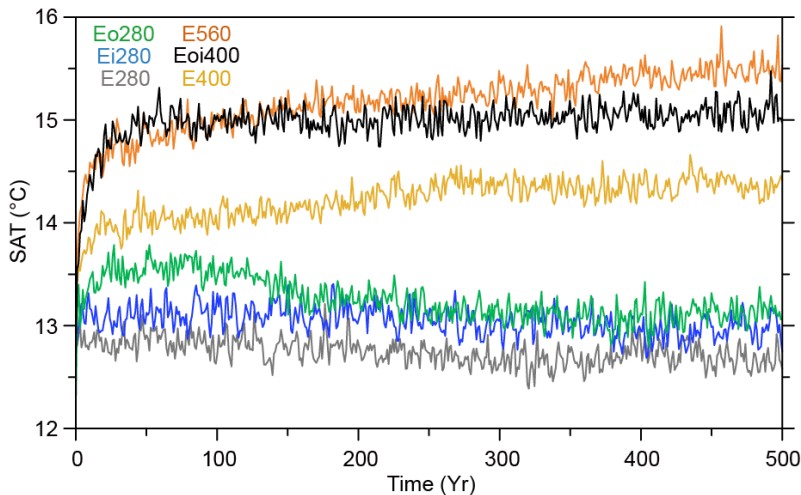

**Figure 4: Time evolution of annual-mean global-mean surface air temperature (SAT; °C) in each run.**





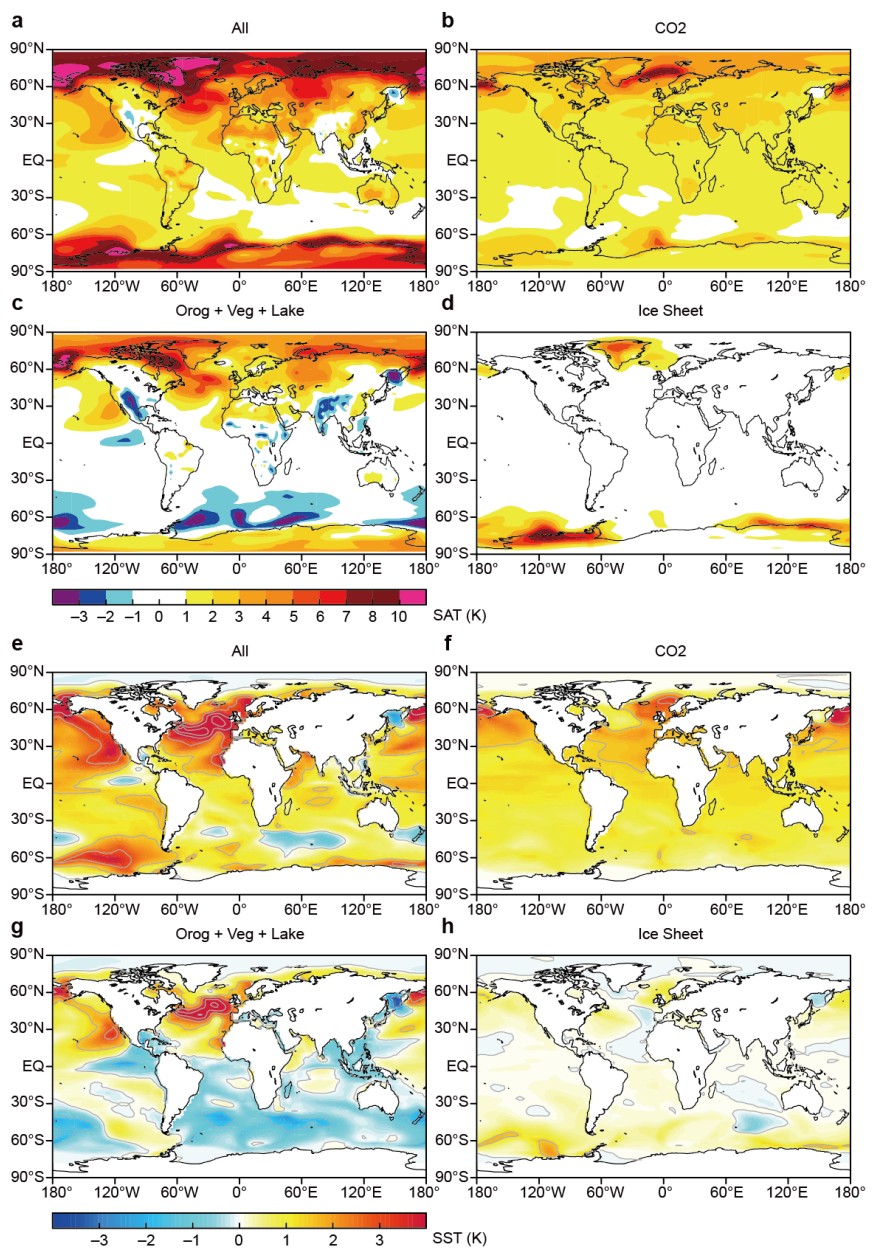



**Figure 5: (a) Anomaly in SAT (K) in Pliocene run relative to pre-industrial run (hereafter *All*). (b) E400 run minus pre-industrial run (*CO2*), (c) Eo280 run minus pre-industrial run (*OVL*), and (d) Ei280 run minus pre-industrial run (*Ice Sheet*), respectively. The anomalies are calculated by averages for the last 50-yr of the individual runs. (e–h) Similar to (a–d), but for sea surface temperature (SST; K). Intervals of grey contours are 1.5 K.**

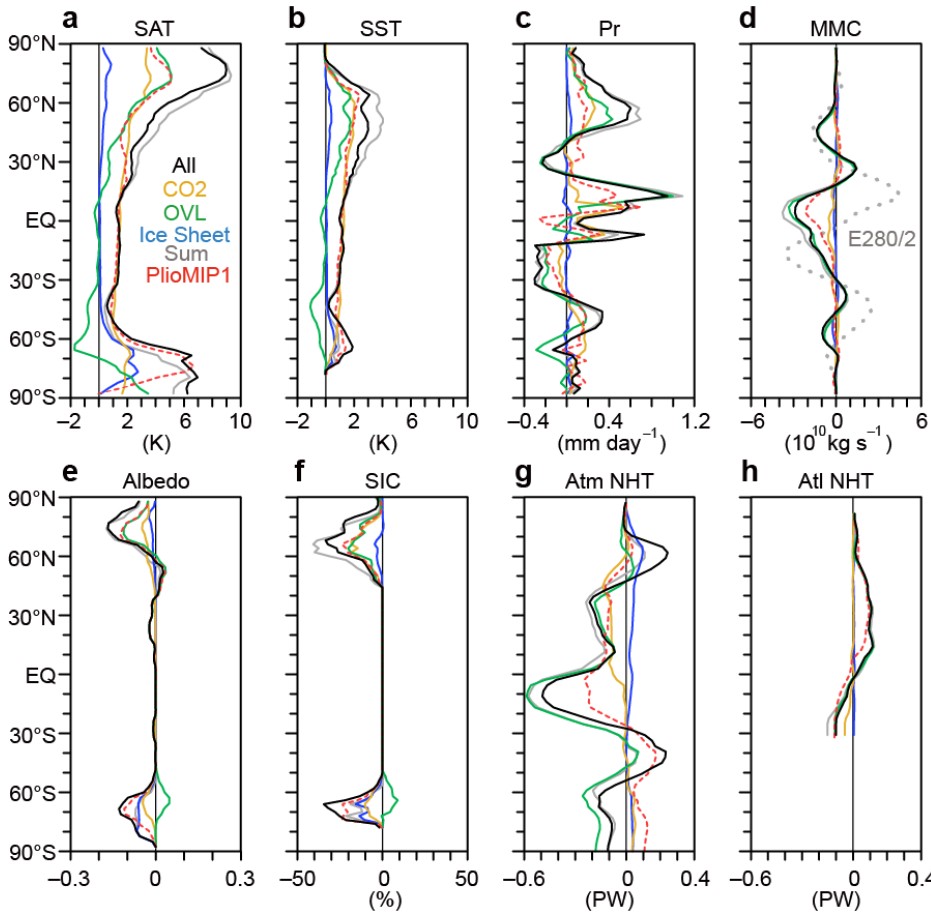

**Figure 6: Zonal-mean anomalies in (a) SAT (K), (b) SST (K), (c) precipitation (mm day⁻¹), (d) mass stream function of mean meridional circulation at 500 hPa level ($10^{10}$ kg s⁻¹), (e) surface albedo, (f) sea ice concentration (%), (g) northward heat transport**
10 **due to atmosphere (PW) and (h) the Atlantic (PW). Black, yellow, green, and blue lines represent *All*, *CO2*, *OVL*, and *Ice Sheet*, respectively. Grey line represents *Sum*. Dashed red lines represent results of AOGCM_NFA run conducted in the identical model in PlioMIP1 (Kamae and Ueda, 2012). Dotted grey line in (g) represents climatology in pre-industrial run multiplied by 0.5.**



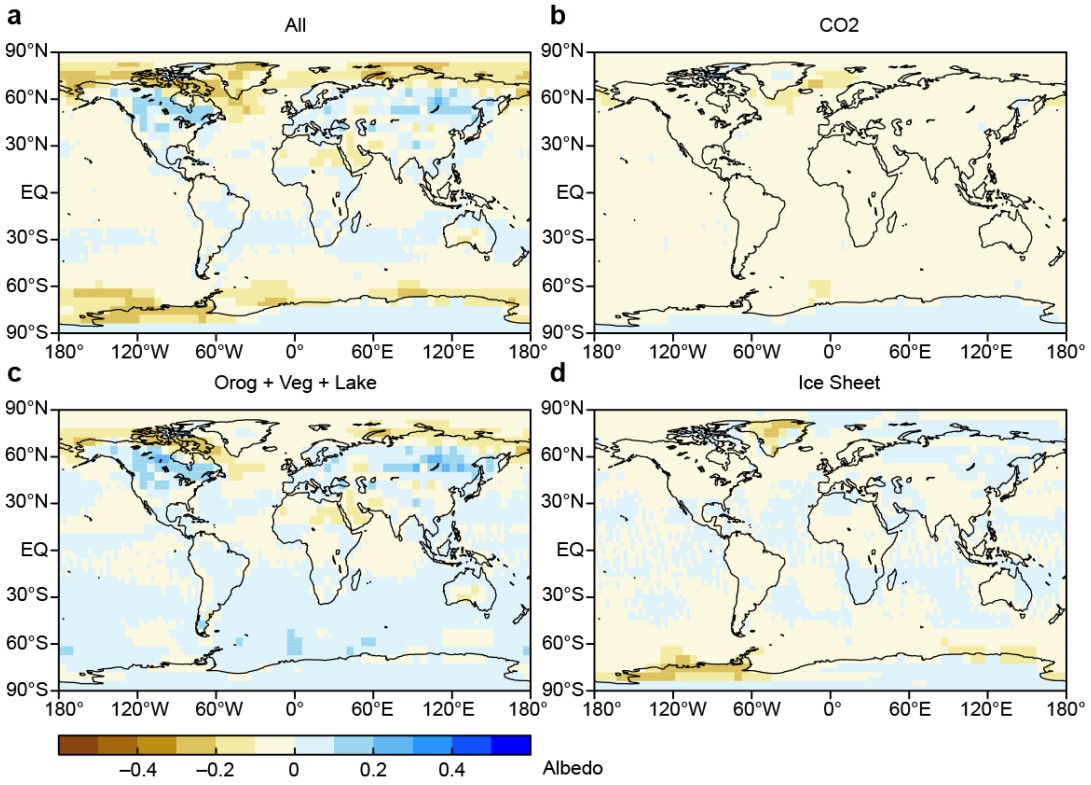

Figure 7: Similar to Fig. 5, but for surface albedo.





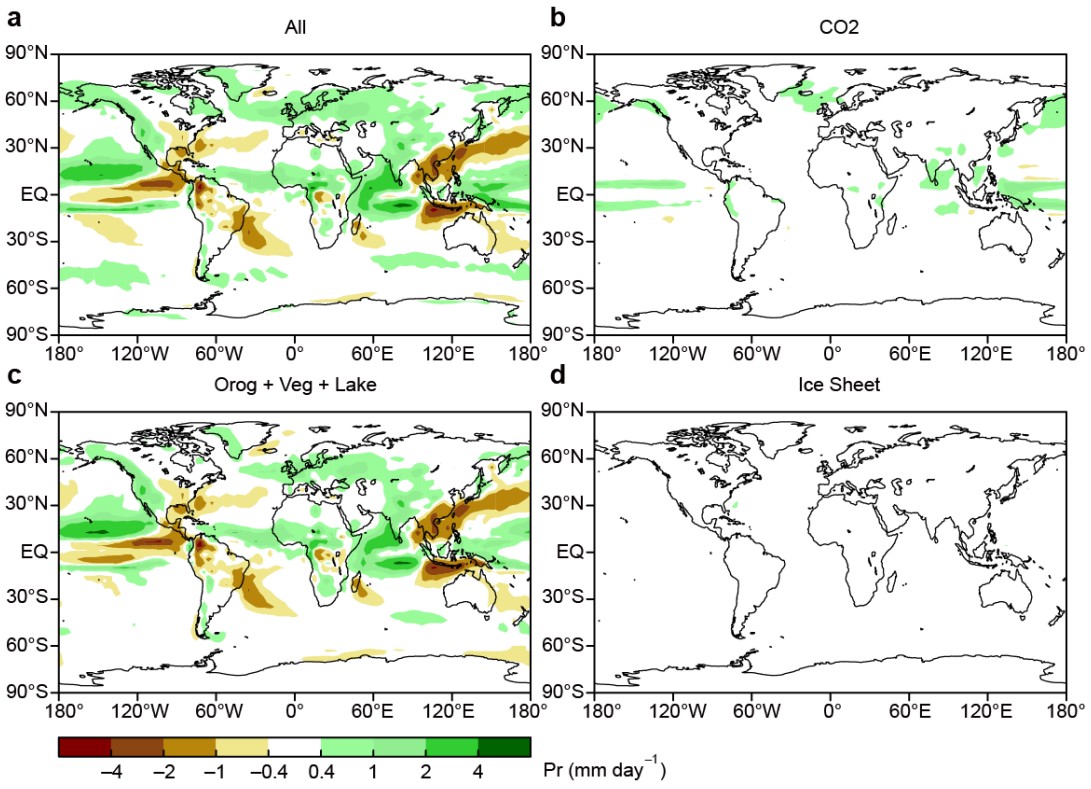

Figure 8: Similar to Fig. 5, but for precipitation (mm day⁻¹).



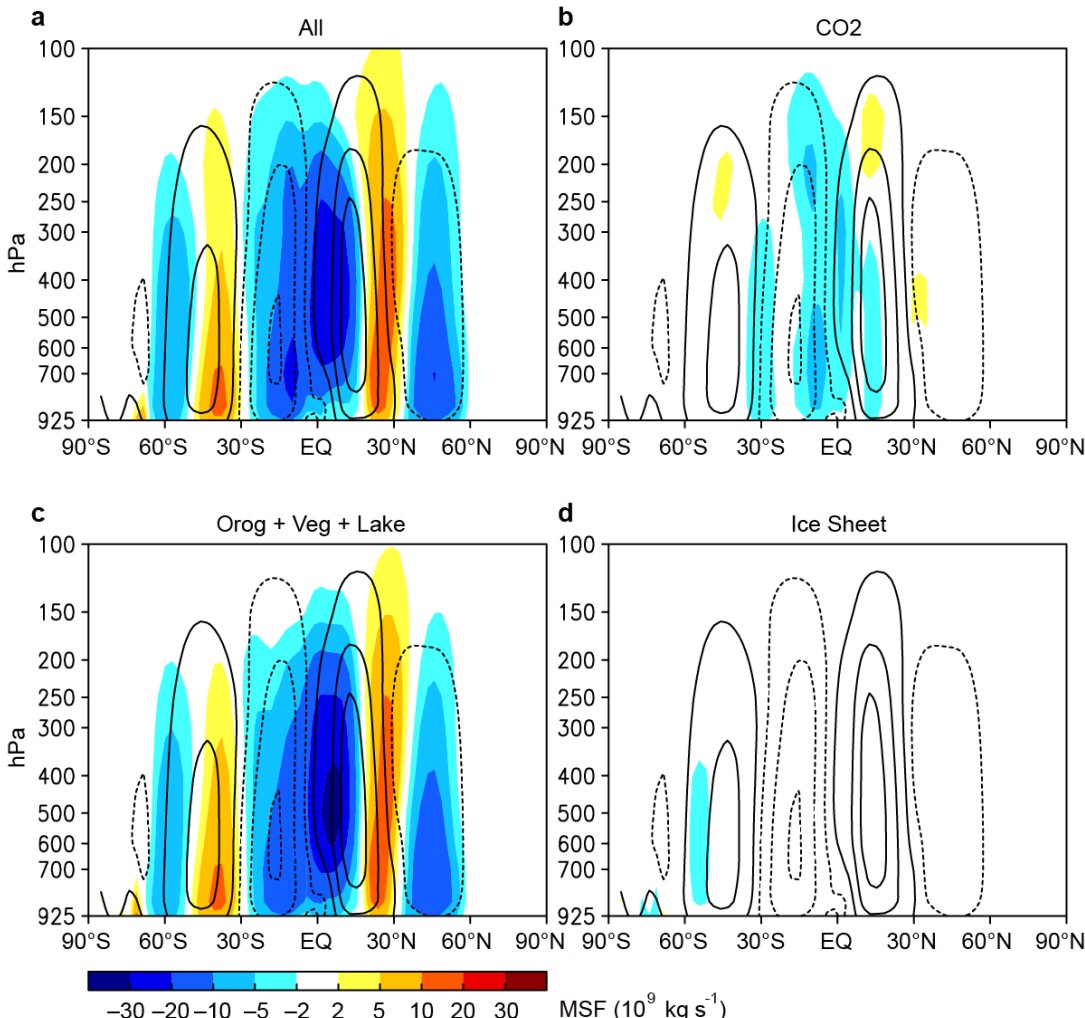

**Figure 9: Anomalies in mass stream function of mean meridional circulation ($10^9$ kg s$^{-1}$). (a) *All* (shading). Contours represent climatological mass stream function in pre-industrial run (±10, 40, 70 $10^9$ kg s$^{-1}$). Solid and dashed contours represent positive and negative anomalies, respectively. (b) *CO2*, (c) *OVL*, and (d) *Ice Sheet*, respectively.**





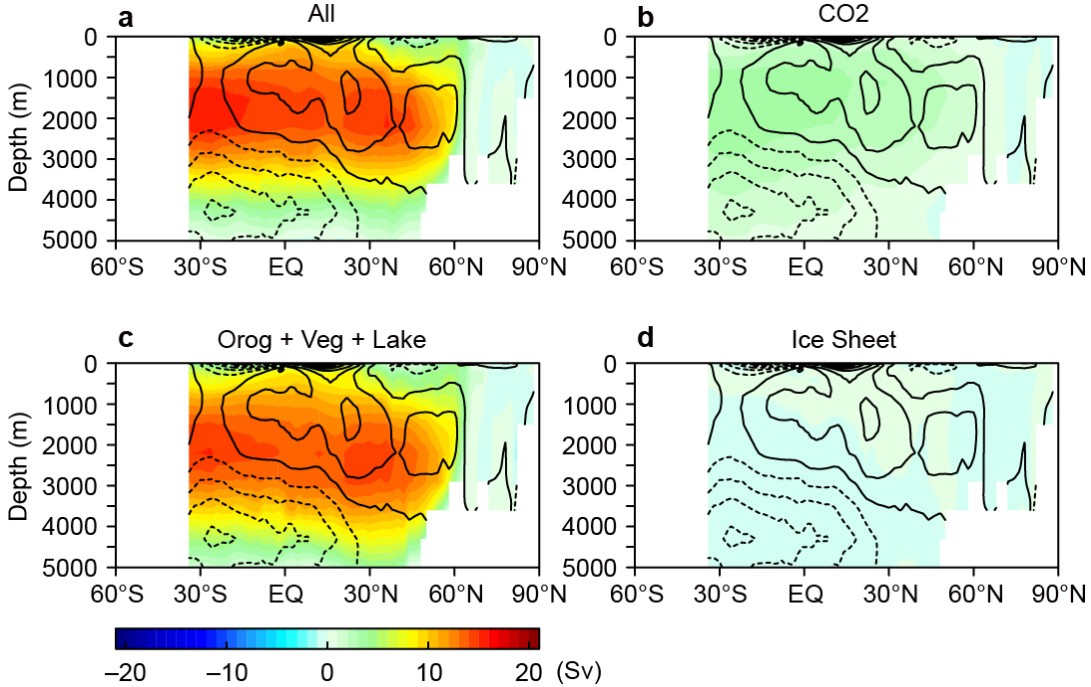

**Figure 10: Anomalies in Atlantic meridional overturning circulation (AMOC; shading; Sv). Contours represent climatological overturning circulation in pre-industrial run (±0, 2, 4, 6, 8 Sv). (a)** *All*, **(b)** *CO2*, **(c)** *OVL*, **and (d)** *Ice Sheet*, **respectively.**





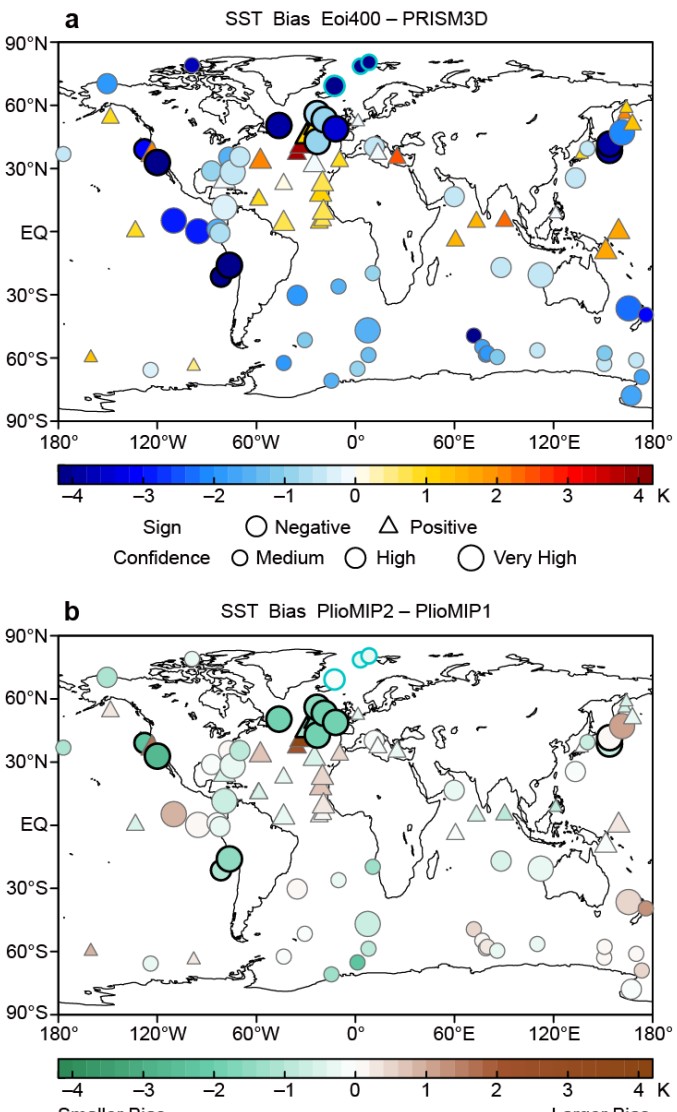

**Figure 11: (a) SST bias (K) in PlioMIP2 Pliocene run compared with PRISM3D proxy-based SST reconstruction (Dowsett et al., 2009). Coloured circles and triangles represent cool and warm biases, respectively. Sizes of plots indicate confidence levels of proxy-based SST estimate (derived from Dowsett et al., 2013). Thick black and blue open circles indicate proxy sites in which estimated SST anomalies are large (between 4.0 and 8.9 K) and extremely large (> 8.9 K), respectively. (b) Comparison of SST biases in Pliocene runs between PlioMIP2 and 1. Coloured plots indicate differences in absolute biases between the two.**





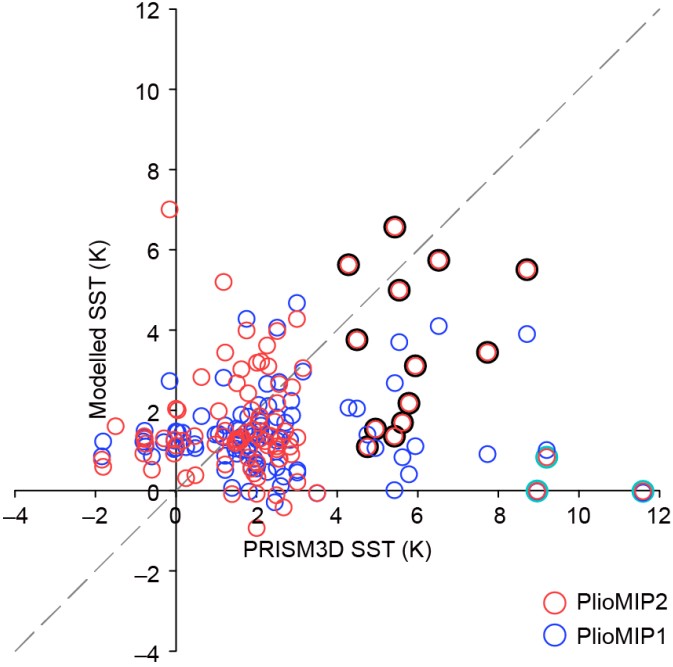

**Figure 12: Scatter diagram of proxy-based SST reconstruction (K) and simulated SST (K). Red and blue circles represent Pliocene runs in PlioMIP2 and 1, respectively. Thick black and blue open circles are identical to Fig. 11. Dashed grey line represents one-by-one line.**

