# Peer review of "Sensitivity of Pliocene climate simulations in MRI-CGCM2.3 to respective boundary conditions"

_Climate of the Past, 2016_

## Referee Comment (RC1) · Anonymous Referee #1 · 6 Jun 2016

This manuscript addresses the pertinent question of what is the sensitivity of the climate to Pliocene boundary conditions, which is an area of research relevant to the scope of Climate of the Past. The title clearly reflects the research study and contents. The paper aims to bridge the gap in understanding of how the climate responds to Pliocene boundary conditions by performing a thorough model analysis using recently revised boundary data. It's a significant contribution to the Pliocene Model Intercomparison Project phase 2 (PlioMIP2), where other models also test this sensitivity. The literature review is comprehensive and clearly identifies the current state of the science. The manuscript contents are organized in a logical manner and the novelty of the research (i.e., newly updated boundary conditions) is clearly described in the introduction. The experiment is well designed and sufficient to answer the questions posed in the manuscript. By using the same version of the model and only changing

the boundary conditions, it allows for an adequate analysis of the sensitivity of the data on climate. The amount of ensembles and spin up equilibrium time of the model is sufficient for the robustness of the results, and the adherence to the PlioMIP2 protocol allows for an future model intercomparison. The model results are presented with an analysis of the mechanisms driving them, and the conclusions drawn are consistent with the interpretation of the results. While the conclusions are substantial, the research requires additional analysis to better explain the responsible physical mechanisms. However, this is clearly identified in the manuscript and may be sufficiently covered in another manuscript. Furthermore, the use of a higher resolution model or earth system model with dynamic vegetation would be a very useful companion study to further evaluate the interactions and feedbacks in the climate system. Overall, the quality of the paper is good and recommended for publication with minor issues addressed below. 1. The color scheme used in Figure 1 (page 19) to illustrate the prescribed land cover for the modern and Pliocene periods makes it hard to distinguish between certain types. For example, deciduous broadleaf+evergreen conifer (03), tundra (10) and land ice (13) are too similar in color and difficult to decipher. A broader range of color scheme is recommended for this figure. 2. The color scheme for Figure 2C (page 20) also makes it difficult to see the single grid cell light blue pixels, in particular, in North America and Asia. I recommend either contouring the small lake areas or using a bolder color to enhance them. 3. For consistency, I recommend using the same surface air temperature (SAT) units throughout the figures and manuscript. For example, Figure 4 uses °C for SAT while Figure 5 uses K. 4. To better follow the naming convention, I recommend reordering the text on page 4 line 30 to be consistent with the "OVL" acronym (e.g., "orography, vegetation, and lakes"). 5. Since "OVL" is used throughout the text, I recommend adding it to the label ("Orog+Veg+Lake (OVL)") for Figures 4C, 4G (page 22), Figure 6C (page 24), Figure 7C (page 25), Figure 8C (page 26), and Figure 9C (page 27).
* * *

---

## Referee Comment (RC2) · Anonymous Referee #2 · 14 Jun 2016

Based on several experiments performed with the model MRI-CGCM2.3, the authors have analyzed the climate response to the updated boundary conditions of PlioMP2, including the CO2 concentration, the orography+vegetation+lake (OVL) as a whole, and the ice sheets. They have also compared their results with those obtained during PlioMP1 by the same model and with proxy data. This work is helpful for understanding better the origin of the Pliocene warm climate and the individual contributions of CO2, OVL and ice sheets. The manuscript is well organized and the introduction and method are well explained. The research topic is quite suitable for publication in Climate of the Past. However, this manuscript is lack of depth due to insufficient explanation on mechanisms. I would recommend to reinforce the analysis and discussion on physical mechanisms before publication by taking into account my following comments.

1. The authors have described the impact of OVL and ice sheets on sea surface

temperature, sea ice, AMOC and the Hadley circulation, but almost no explanation is given on the physical mechanisms. I would recommend to add explanations on how the changes in OVL and ice sheets cause the changes in these climatic variables.

2. The OVL is the major contributor for a stronger AMOC. What is the mechanism?

3. Page 8, line 29: how do the ice sheet and OVL enhance northward heat transport?

4. The OVL causes significant change in the tropical precipitation. Any idea about which factor contributes the most, orography, vegetation or lake?

5. In the ice sheet experiment, how is the ice sheet defined? By changes in albedo and topography? In the lake experiment, how is the lake defined in the model? These should be explicitly explained in the paper.

6. The effect of ice sheets should include the effect of its topography. In the OVL experiment, the effect of the ice sheet topography seems to be also included. In this case, there should be an overlap of the effect of ice sheet topography in these two experiments. Is it true?

7. Page 9, line 19: Please explain what the "nonlinear residual" means in terms of physics.

8. Page 5, line 10: would there be any difference between with and without the addition of deep ocean temperature to the initial condition?

9. Page 5, line 26: what does the "nonlinear" mean exactly?

10. Page 5, line 29: does it suggest no interactive effect of the CO2, OVL and ice sheets?

11. Page 8, line 12: isn't better to change "suggesting" to "resulting from"?

12. Fig11: how was the confidence level (medium, high, very high) defined?

13. Page 10: lines 2-3: PlioMP2 is cooler than PlioMP1 over N. Atlantic, so the underestimation of the warming there is actually increased but not reduced as written by the authors. And why A.Atlantic is cooler in PlioMP2 than PlioMP1?

14. Page 10, line 15: the linear additivity of the Pliocene climate simulation is not necessarily obvious at regional scale (see fig6).

15. Page 10, line 21: Please comment what are the possible reasons that the model fails to reproduce the extremely warm condition over the Arctic to high-latitude North Atlantic region.

---

## Author Comment (AC1) · 17 Jun 2016

We appreciate your careful reading of our manuscript and your helpful suggestions for its improvement. I'm happy to incorporate all of your suggestions to our manuscript.

> 1. The color scheme used in Figure 1 (page 19) to illustrate the prescribed land cover for the modern and Pliocene periods makes it hard to distinguish between certain types. For example, deciduous broadleaf+evergreen conifer (03), tundra (10) and land ice (13) are too similar in color and difficult to decipher. A broader range of color scheme is recommended for this figure. > 2. The color scheme for Figure 2C (page 20) also makes it difficult to see the single grid cell light blue pixels, in particular, in North America and Asia. I recommend either contouring the small lake areas or using a bolder color to enhance them.

[Figure]

We will improve color schemes in Figures 1 and 2c in revised version of our manuscript for clarity.

> 3. For consistency, I recommend using the same surface air temperature (SAT) units throughout the figures and manuscript. For example, Figure 4 uses °C for SAT while Figure 5 uses K.

In the revised manuscript, we will use "°C" consistently in main text and Figures 4-6, 11-12 and Tables according to your suggestion.

> 4. To better follow the naming convention, I recommend reordering the text on page 4 line 30 to be consistent with the "OVL" acronym (e.g., "orography, vegetation, and lakes"). > 5. Since "OVL" is used throughout the text, I recommend adding it to the label ("Orog+Veg+Lake (OVL)") for Figures 4C, 4G (page 22), Figure 6C (page 24), Figure 7C (page 25), Figure 8C (page 26), and Figure 9C (page 27).

We will modify the order of the words in main text, and use "OVL" consistently in figure labels. Thank you for your kind suggestions.

---

## Author Comment (AC2) · 28 Jun 2016

> Based on several experiments performed with the model MRI-CGCM2.3, the authors have analyzed the climate response to the updated boundary conditions of PlioMP2, including the CO2 concentration, the orography+vegetation+lake (OVL) as a whole, and the ice sheets. They have also compared their results with those obtained during PlioMP1 by the same model and with proxy data. This work is helpful for understanding better the origin of the Pliocene warm climate and the individual contributions of CO2, OVL and ice sheets. The manuscript is well organized and the introduction and method are well explained. The research topic is quite suitable for publication in Climate of the Past. However, this manuscript is lack of depth due to insufficient explanation on mechanisms. I would recommend to reinforce the analysis and discussion on physical mechanisms before publication by taking into account my following comments.

[Figure]

Thank you for your careful reading and many constructive suggestions. Firstly ice sheet effect is estimated by changing (1) land cover and (2) topography. There are no overlaps between estimated OVL and Ice Sheet effects, so one of the reviewer's concerns (comments #5 and #6) is not true in this case.

According to reviewer's comments, we add discussions on physical processes responsible for the derived climate anomalies in Eoi400 (Pliocene) run and sensitivity runs. In revised version of our manuscript, we clarify physical meaning of "nonlinear" residual term estimated in this study. Responses to individual comments are listed below.

> 1. The authors have described the impact of OVL and ice sheets on sea surface temperature, sea ice, AMOC and the Hadley circulation, but almost no explanation is given on the physical mechanisms. I would recommend to add explanations on how the changes in OVL and ice sheets cause the changes in these climatic variables.

We agree to the comment. In our original manuscript, we only introduced general characteristics and individual roles of boundary forcings in climate responses to the Pliocene boundary conditions.

In response to the ice sheet reduction, land surface warms up locally (Fig. 5a) due to lower albedo (Fig. 7d) and lower orography (Fig. 3). The regional warming may result in sea ice reduction and sea surface warming over the surrounding regions (Figs. 5h, 6b, f). Here secondary changes in atmospheric and ocean circulations are generally smaller than the responses to other forcings (Figs. 6d, g, h, 9d, 10d). The change in atmospheric northward heat transport (Fig. 6g) may be related to mid-latitude atmospheric eddy (due to orography change and/or change in meridional temperature gradient in the troposphere and/or other processes), but more detailed analyses are needed to examine quantitatively physical processes contributing to the meridional heat transport.

OVL effect contains variety of forcing-feedback processes: direct influences of altered orography, vegetation, lakes and secondary influences/feedbacks initiated by them.

[Figure]

For vegetation, previous studies pointed out that vegetation cover change can result in local and global climate through biogeophysical (albedo and evapotranspiration; Davin and de Noblet-Ducoudré 2010; Willeit et al. 2013; Zhang and Jiang 2014) effect (and biogeochemistry effect; but this is not considered in this study because the model does not treat carbon cycle). Generally, large-scale vegetation change over the high latitude has substantial impact on temperature through change in albedo (Fig. 1; Davin and de Noblet-Ducoudré 2010). Decrease in albedo over the Northern Hemisphere high latitude due to northward shift of boreal forest (Fig. 1) can be a trigger for climate feedback (e.g. ice albedo feedback including sea ice reduction; Fig. 6f) and Arctic warming amplification (Fig. 6a), similar to previous studies (e.g. Zhang and Jiang 2014). The resultant meridional warming gradient induced by OVL effect (discussed in Sect. 4) can influence on global atmospheric circulation including strength and position of the mean meridional circulation including Hadley circulation (Figs. 6d, 9c), mid-latitude synoptic eddies, and associated atmospheric heat transport (Fig. 6g; see response to comment #3), precipitation pattern (Figs. 6c, 8c; see response to comment #4). Change in sea water density flux over the North Atlantic, a driving factor for the AMOC, can be drastically changed in response to OVL forcing via surface latent and sensible heat flux, radiative (longwave and shortwave) flux, and salinity budget (see response to comment #2). Physical processes responsible for the AMOC change simulated in Eoi400 and Eo400 are examined in detail in a separated paper.

In revised version of our manuscript, we add explanations and discussions on the derived climate responses to the boundary forcings to the main text.

> 2. The OVL is the major contributor for a stronger AMOC. What is the mechanism?

Change in sea surface density flux can be one of the controlling factors for the change in AMOC (Speer and Tziperman, 1992). Warmer surface air and sea surface water, changes in humidity and cloud cover can influence on surface sensible flux, latent heat flux, and longwave and shortwave radiation at sea surface over the North Atlantic. In addition, changes in river runoff, sea ice melt, and precipitation minus evaporation can

affect sea surface salinity over this region. Such heat and salinity forcings can result in stronger/weaker AMOC in a perturbed climate (Speer and Tziperman, 1992). More detailed analyses will be conducted and reported in a separated paper, as described in the original manuscript. In revised version of our manuscript, we add possible factors for the AMOC change due to OVL forcing to the main text.

> 3. Page 8, line 29: how do the ice sheet and OVL enhance northward heat transport?

The anomalous atmospheric northward heat transport in Ice Sheet effect and OVL effect are found over 50N-70N. Here these anomalous transports are not attributed to zonal-mean meridional circulation because their changes are quite small (Fig. 6d). Here the mid-latitude eddy transport may play an important role in this anomalous heat transport. Ulbrich et al. (2009) summarized physical processes responsible for change in mid-latitude eddy activity in projected future climate. They revealed that changes in meridional temperature gradient in the middle-upper troposphere and near the surface can result in changes in jet stream and the mid-latitude eddies. Li et al. (2015) showed substantial changes in meridional temperature gradient in the troposphere in selected PlioMIP1 models. Actually, tropospheric meridional temperature gradient shows a similar anomalous pattern to Li et al. (2015). Such changes in meridional temperature gradient (Fig. 6a) can be one of factors for the anomalous meridional heat transport due to the atmosphere. In addition, anomalous orography over Greenland as a part of Ice Sheet effect may also contribute to changes in jet stream and mid-latitude heat transport. We add discussion on the physical processes associated with the enhanced heat transport.

> 4. The OVL causes significant change in the tropical precipitation. Any idea about which factor contributes the most, orography, vegetation or lake?

Willeit et al. (2013) and Zhang and Jiang (2014) also showed that change in vegetation cover from pre-industrial to Pliocene condition can result in significant change in tropical precipitation. Albedo reduction over the Northern Hemisphere high latitude due to

vegetation change (e.g. northward shift of boreal forest; Figs. 1, 6e, 7c) contributes to the Arctic warming amplification (Figs. 5c, 6a). Associated feedback processes (e.g. snow and sea ice) amplify the high latitude warming, leading to the substantial change in meridional warming gradient (discussed in Sect. 4). Resultant warming gradient between the Northern/Southern Hemispheres (found in OVL effect; Fig. 6a) can lead to change in strength and position of the intertropical convergence zone (ITCZ)-related precipitation (e.g. Zhang and Delworth 2006; Braconnot et al. 2007) including the increase in tropical North Atlantic-North African precipitation (Fig. 8c). Other processes (evaporation change over tropical and subtropical land due to changes in vegetation and lake, orography-induced atmospheric circulation changes) may also play roles in the tropical precipitation change. We add discussion on the physical processes above to the main text.

From a regional perspective, tropical precipitation changes in the Pliocene run may strongly be related to changes in individual monsoon systems. In the PlioMIP2, systematic investigations of regional monsoon behaviors during the late Pliocene are planned to conduct. Detailed physical processes (e.g. Zhang R. et al. 2013) will be examined in future studies.

> 5. In the ice sheet experiment, how is the ice sheet defined? By changes in albedo and topography? In the lake experiment, how is the lake defined in the model? These should be explicitly explained in the paper.

> 6. The effect of ice sheets should include the effect of its topography. In the OVL experiment, the effect of the ice sheet topography seems to be also included. In this case, there should be an overlap of the effect of ice sheet topography in these two experiments. Is it true?

In this study, we estimated the ice sheet effect by comparing results of simulations with and without the ice sheet over the part of Greenland and Antarctic Continent (Figs. 1, 3). Here the anomalous ice sheet was prescribed as anomalous orography and land

cover, so the resultant ice sheet effect contains both orography and land cover effects. The derived ice sheet effect is not overlapped with OVL effect because "orography effect" in OVL does not include the ice sheet orography.

In our original manuscript, details of the prescribed boundary conditions were not sufficiently explained. We revise the explanations on the experimental setups.

The lakes are treated in land surface model as inland water in grid boxes with a drainage basin unconnected to oceans (Yukimoto et al. 2006). The subgrid lake parameterization included in land surface model predicts water budget, but the lake surface temperature is predicted by the heat budget at the water surface, assuming a slab with a thickness of 50m, as described in Sect. 2.1. In the control run, five lakes were modeled as inland waters (Fig. 2a, Yukimoto et al. 2006). In the Pliocene run, additional lakes are implemented to land surface (Fig. 2b). We add explanations of the treatment of lakes to Sect. 2.1.

> 7. Page 9, line 19: Please explain what the "nonlinear residual" means in terms of physics.

The nonlinear residual term defined as Eq. 6 in this study means a departure from 'linear additivity of forcing-response relationships' (e.g. Shiogama et al. 2013). If external forcing A and B were added to system separately or together, climate response to "A+B" forcing is not necessarily identical to sum of climate responses in A forcing run and B forcing run. In case for discussion in Page 9 line 19, sea ice reduction and surface warming are larger in Sum than All (Fig. 6f). Over the regions with Arctic sea ice edge, climatological sea ice concentration is limited. Sum of sea ice reductions due to strong forcings A and B (for example, 12% and 8%) can be larger than climatological sea ice concentration (for example, 15%), resulting in nonlinear forcing-sea ice reduction relationship (-20% is larger reduction than -15%) because sea ice concentration is always larger than or equal to 0%. We add explanation on possible physical processes responsible for the nonlinear residual term discussed here.

> 8. Page 5, line 10: would there be any difference between with and without the addition of deep ocean temperature to the initial condition?

The anomalous deep ocean temperature imposed in PlioMIP1 run may play some roles, but general ocean warming pattern and AMOC change are generally similar between PlioMIP2 and PlioMIP1 (Figs. 5e, 6h in this paper, and Fig. 8e in Kamae and Ueda 2012), suggesting that the imposed deep ocean temperature is not the dominant contributor for the simulated Pliocene climate anomaly, at least after the long-term integrations (500 years). We add a note on influence of the imposed deep ocean temperature as an initial condition.

> 9. Page 5, line 26: what does the "nonlinear" mean exactly?

The "nonlinear" effect stated here means nonlinear forcing-feedback relationship. Please see response to comment #7. We add meaning of "nonlinear" effect to Sect. 2.3.

> 10. Page 5, line 29: does it suggest no interactive effect of the CO2, OVL and ice sheets?

The interactive effect suggested by Residual term is minor to All, but the residual term is not negligible for regional climate anomalies. We revise this part to clarify that Residual can be found in regional scale.

> 11. Page 8, line 12: isn't better to change "suggesting" to "resulting from"?

We revise as "resulting from" according to your comment.

> 12. Fig11: how was the confidence level (medium, high, very high) defined?

The confidence level shown here was derived from Dowsett et al. (2013). In their paper, confidence level of proxy-suggested climate anomaly was defined by chronology, sampling density, sampling quality and performance of quantitative method. Details of confidence scheme used in the PRISM SST reconstruction can be found in Dowsett

et al. (2013). We add "defined by chronology, sampling density, sampling quality and performance of quantitative method" to the caption.

> 13. Page 10: lines 2-3: PlioMP2 is cooler than PlioMP1 over N. Atlantic, so the underestimation of the warming there is actually increased but not reduced as written by the authors. And why A.Atlantic is cooler in PlioMP2 than PlioMP1?

We guess this is a reviewer's misunderstanding on the meaning of Fig. 11. SST in PlioMIP2 run is warmer than PlioMIP1 run (Figs. 6b, 11a), so the underestimation of the warming is reduced (Fig. 11b). Generally PlioMIP2 run shows larger warming than PlioMIP1 run (Table 3). Both underestimation of warming over the North and South Atlantic middle and high latitude (Fig. 11a) are reduced in the PlioMIP2 run (Fig. 11b). We revise main text in section 5 to avoid confusing.

> 14. Page 10, line 15: the linear additivity of the Pliocene climate simulation is not necessarily obvious at regional scale (see fig6).

We agree to your comment. Global-mean response shows good linear additivity (Table 3), but the linear additivity is limited for the regional climate responses. We add a note to this section as follows: "However, linear additivity does not hold so well for regional climate responses including sea ice reduction over the high latitude oceans."

> 15. Page 10, line 21: Please comment what are the possible reasons that the model fails to reproduce the extremely warm condition over the Arctic to high-latitude North Atlantic region.

This underestimation of the high latitude North Atlantic warming was also found in PlioMIP1 multiple climate models. Haywood et al. (2013) pointed out that almost all the PlioMIP1 AOGCMs underestimated this extreme warming. Possible reasons why the model cannot reproduce the extremely warm condition over the high latitude North Atlantic may be associated with (1) AMOC biases in models, (2) sea ice biases in models, and/or (3) uncertainty in SST estimate based on proxy records (Robinson 2009;

Haywood et al. 2013). Haywood et al. (2013) noted that this large model/data discord was highly dependent on geochemically-based proxy mean annual temperature estimate and was not derived from faunal based estimates of cold/warm month means. They suggested that we should not rely on this model/data discord too much until more variety of proxy records is available from more locations in the high latitude North Atlantic and the Arctic. We add discussion on this issue to section 5.

Additional references:

Braconnot, P., Otto-Bliesner, B., Harrison, S., Joussaume, S., Peterchmitt, J.-Y., Abe-Ouchi, A., Crucifix, M., Driesschaert, E., Fichefet, Th., Hewitt, C. D., Kageyama, M., Kitoh, A., Loutre, M.-F., Marti, O., Merkel, U., Ramstein, G., Valdes, P., Weber, L., Yu, Y., and Zhao, Y.: Results of PMIP2 coupled simulations of the Mid-Holocene and Last Glacial Maximum – Part 2: feedbacks with emphasis on the location of the ITCZ and mid- and high latitudes heat budget, Clim. Past, 3, 279-296, doi:10.5194/cp-3-279-2007, 2007.

Davin, E. L. and de Noblet-Ducoudré, N.: Climatic impact of global-scale deforestation: Radiative versus nonradiative processes, J. Clim., 23, 97–112, 2010.

Shiogama, H., Stone, D. A., Nagashima, T., Nozawa, T., and Emori, S.: On the linear additivity of climate forcing-response relationships at global and continental scales, Int. J. Climatol., 33, 2542–2550, 2013.

Speer, K., and Tziperman, E.: Rates of water mass formation in the North Atlantic Ocean, J. Phys. Oceanogr., 22, 93–104, 1992.

Ulbrich, U., Leckebusch, G. C., and Pinto, J. G.: Extra-tropical cyclones in the present and future climate: A review, Theor. Appl. Climatol., 96, 117–131, 2009.

Willeit, M., Ganopolski, A., and Feulner, G.: On the effect of orbital forcing on mid-Pliocene climate, vegetation and ice sheets, Clim. Past, 9, 1749–1759, 2013.

Zhang, R., and Jiang, D.: Impact of vegetation feedback on the mid-Pliocene warm

climate, Adv. Atmos. Sci., 31, 1407–1416, 2014.

---

## Author Response (AR1)

**#cp-2016-50**

**Sensitivity of Pliocene climate simulations in MRI-CGCM2.3 to respective boundary conditions**

by Youichi Kamae, Kohei Yoshida, and Hiroaki Ueda

**General response to reviewers**

We appreciate reviewers' careful reading of our manuscript and their helpful suggestions for its improvement. We have carefully considered their comments in order to improve our manuscript. The revised parts are highlighted in blue in the marked-up manuscript (attached below). Major points of the current revision are summarized as follows.

According to reviewer's comment, we incorporate additional discussions on physical processes responsible for the simulated Pliocene climate anomalies (e.g. precipitation, overturning circulation, and heat transport) and roles of individual boundary conditions. By comparing imposed forcings and associated feedbacks (e.g. orography and albedo) and changes in thermodynamic and dynamic atmospheric and oceanic properties (e.g. meridional temperature gradient, overturning circulation) in the individual sensitivity experiments and comparing them with previous modelling studies, possible physical mechanisms can be discussed. Additional discussions facilitate further systematic, process-based evaluations of the Pliocene AMOC, monsoons and atmospheric circulations, that are planned to conduct under the PlioMIP2 framework.

We also clarify experimental setups and diagnosis method used in this study, according to reviewers' comments. We also improve colour schemes, titles and captions of figures. For details, please see response to individual comments listed below.

**Reply to comments by the reviewer #1**

We thank the reviewer for his/her careful reading of the manuscript. The manuscript has been revised following the review comments (*Italic fonts* are the review comments and blue fonts are item-by-item responses).

*Reviewer #1:*

*This manuscript addresses the pertinent question of what is the sensitivity of the climate to Pliocene boundary conditions, which is an area of research relevant to the scope of Climate of the Past. The title clearly reflects the research study and contents. The paper aims to bridge the gap in understanding of how the climate responds to Pliocene boundary conditions by performing a thorough model analysis using recently revised boundary data. It's a significant contribution to the Pliocene Model Intercomparison Project phase 2 (PlioMIP2), where other models also test this sensitivity. The literature review is comprehensive and clearly identifies the current state of the science. The manuscript contents are organized in a logical manner and the novelty of the research (i.e., newly updated boundary conditions) is clearly described in the introduction. The experiment is well designed and sufficient to answer the questions posed in the manuscript. By using the same version of the model and only changing the boundary conditions, it allows for an adequate analysis of the sensitivity of the data on climate. The amount of ensembles and spin up equilibrium time of the model is sufficient for the robustness of the results, and the adherence to the PlioMIP2 protocol allows for an future model intercomparison. The model results are presented with an analysis of the mechanisms driving them, and the conclusions drawn are consistent with the interpretation of the results. While the conclusions are substantial, the research requires additional analysis to better explain the responsible physical mechanisms. However, this is clearly identified in the manuscript and may be sufficiently covered in another manuscript. Furthermore, the use of a higher resolution model or earth system model with dynamic vegetation would be a very useful companion study to further evaluate the interactions and feedbacks in the climate system. Overall, the quality of the paper is good and recommended for publication with minor issues addressed below.*

We appreciate your careful reading of our manuscript and your helpful suggestions for its improvement. I'm happy to incorporate all of your suggestions to our manuscript.

*1. The color scheme used in Figure 1 (page 19) to illustrate the prescribed land cover for the modern and Pliocene periods makes it hard to distinguish between certain types. For example, deciduous broadleaf+evergreen conifer (03), tundra (10) and land ice (13) are too similar in color and difficult to decipher. A broader range of color scheme is recommended for this figure.*

*2. The color scheme for Figure 2C (page 20) also makes it difficult to see the single grid cell light blue pixels, in particular, in North America and Asia. I recommend either contouring the small lake areas or using a bolder color to enhance them.*

We improve colour schemes in Figures 1 and 2c in the revised version of our manuscript for clarity: i.e. we modify colours for land ice (grey) and tundra (aquamarine; Fig. 1), and colour for no-lake regions (dark grey; Fig. 2c).

[Figure]

Figure 1: Prescribed land cover (SiB2 classification) for (a) modern and (b) Pliocene conditions.

[Figure]

Figure 2c: Prescribed lake area fraction over land. Pliocene minus modern.

*3. For consistency, I recommend using the same surface air temperature (SAT) units throughout the figures and manuscript. For example, Figure 4 uses C for SAT while Figure 5 uses K.*

In our revised manuscript, we use "°C" consistently in main text and Figures 4-6, 11-12 and Tables according to your suggestion. For reference, the revised parts are highlighted in blue in the mark-up manuscript (attached).

*4. To better follow the naming convention, I recommend reordering the text on page 4 line 30 to be consistent with the "OVL" acronym (e.g., "orography, vegetation, and lakes").*
*5. Since "OVL" is used throughout the text, I recommend adding it to the label ("Orog+Veg+Lake (OVL)") for Figures 4C, 4G (page 22), Figure 6C (page 24), Figure 7C (page 25), Figure 8C (page 26), and Figure 9C (page 27).*

We modify the order of the words in main text, and use "OVL" consistently in figure labels (Figures 5c, g, 6, 7c, 8c, 9c, 10c). Thank you for your kind suggestions.
* * *
The end of reply for #1

**Reply to comments by the reviewer #2**

We thank the reviewer for his/her careful reading of the manuscript. The manuscript has been revised following the review comments (*Italic fonts* are the review comments and blue fonts are item-by-item responses).

*Reviewer #2:*

*Based on several experiments performed with the model MRI-CGCM2.3, the authors have analyzed the climate response to the updated boundary conditions of PlioMP2, including the CO2 concentration, the orography+vegetation+lake (OVL) as a whole, and the ice sheets. They have also compared their results with those obtained during PlioMP1 by the same model and with proxy data. This work is helpful for understanding better the origin of the Pliocene warm climate and the individual contributions of CO2, OVL and ice sheets. The manuscript is well organized and the introduction and method are well explained. The research topic is quite suitable for publication in Climate of the Past. However, this manuscript is lack of depth due to insufficient explanation on mechanisms. I would recommend to reinforce the analysis and discussion on physical mechanisms before publication by taking into account my following comments.*

Thank you for your careful reading and many constructive suggestions. Firstly ice sheet effect is estimated by changing (1) land cover and (2) topography. There are no overlaps between estimated OVL and Ice Sheet effects, so one of the reviewer's concerns (comments #5 and #6) is not true in this case.

According to your comments, we add discussions on physical processes responsible for the derived climate anomalies in Eoi400 (Pliocene) run and sensitivity runs. In the revised version of our manuscript, we clarify physical meaning of "nonlinear" residual term estimated in this study. Responses to individual comments are listed below.

*1. The authors have described the impact of OVL and ice sheets on sea surface temperature, sea ice, AMOC and the Hadley circulation, but almost no explanation is given on the physical mechanisms. I would recommend to add explanations on how the changes in OVL and ice sheets cause the changes in these climatic variables.*

We agree to the comment. In our original manuscript, we only introduced general characteristics and individual roles of boundary forcings in climate responses to the Pliocene boundary conditions.

In response to the ice sheet reduction, land surface warms up locally (Fig. 5d) due to lower

albedo (Fig. 7d) and lower orography (Fig. 3). The regional warming may result in sea ice reduction and sea surface warming over the surrounding regions (Figs. 5h, 6b, f). Here secondary changes in atmospheric and ocean circulations are generally smaller than the responses to other forcings (Figs. 6d, g, h, 9d, 10d). The change in atmospheric northward heat transport (Fig. 6g) may be related to mid-latitude atmospheric eddy (due to changes orography and meridional temperature gradient in the troposphere), but more detailed analyses are needed to examine quantitatively physical processes contributing to the meridional heat transport.

OVL effect contains variety of forcing-feedback processes: direct influences of altered orography, vegetation, lakes and secondary influences/feedbacks initiated by them. For vegetation, previous studies pointed out that change in vegetation cover can result in local and global climate response through biogeophysical (albedo and evapotranspiration; Davin and de Noblet-Ducoudré 2010; Willeit et al. 2013; Zhang and Jiang 2014) effect (and biogeochemistry effect; but this is not considered in this study because the model does not treat carbon cycle). Generally, large-scale vegetation change over the high latitude has substantial impact on temperature through change in albedo (Fig. 1; Davin and de Noblet-Ducoudré 2010). Decrease in albedo over the Northern Hemisphere high latitude due to northward shift of boreal forest (Fig. 1) can be a trigger for climate feedback (e.g. ice albedo feedback including sea ice reduction; Fig. 6f) and Arctic warming amplification (Fig. 6a), similar to previous studies (e.g. Zhang and Jiang 2014). The resultant meridional warming gradient induced by OVL effect (discussed in Sect. 4) can influence on global atmospheric circulation including strength and position of the mean meridional circulation including Hadley circulation (Figs. 6d, 9c), mid-latitude synoptic eddies, and associated atmospheric heat transport (Fig. 6g; see response to comment #3), and precipitation pattern (Figs. 6c, 8c; see response to comment #4). Change in sea water density flux over the North Atlantic, a driving factor for the AMOC, can be drastically changed in response to OVL forcing via surface latent and sensible heat flux, radiative (longwave and shortwave) flux, and salinity budget (see response to comment #2).

In the revised version of our manuscript, we add explanations and discussions on the derived climate responses to the boundary forcings to the main text. In Sect. 3.2, for example, we add explanations on the Northern Hemisphere high latitude warming.

Page 7 line 19: Over the Northern Hemisphere high latitude, the reduced ice sheets (Figs. 1, 3) and a reduction of surface albedo (Figs. 6e, 7c) due to northward shift of boreal forest (deciduous conifer, tundra and bare soil regions over northern Canada and northeastern Eurasia; Fig. 1) result in regional warming (Fig. 5c, d). Surface snow cover (not shown) also affects partly the land surface albedo. The high latitude SAT is more sensitive to imposed albedo change due to the altered vegetation cover

(Fig. 1) than low latitude (Davin and de Noblet-Ducoudré, 2010), consistent with the substantial Arctic warming found in current study (Fig. 5) and previous studies (Willeit et al., 2013; Zhang and Jiang, 2014).

We also revise the manuscript according to your comments below. Please see responses below for details.

*2. The OVL is the major contributor for a stronger AMOC. What is the mechanism?*

Change in sea surface density flux can be one of the contributing factors for the change in AMOC (Speer and Tziperman, 1992). Warmer surface air and sea surface water, changes in humidity, cloud cover, and surface wind speed can influence on surface sensible flux, latent heat flux, and longwave and shortwave radiation at sea surface over the North Atlantic. In addition, changes in river runoff, sea ice melt, and precipitation minus evaporation can affect sea surface salinity over this region. Such heat and salinity forcings can result in stronger/weaker AMOC in a perturbed climate (Speer and Tziperman, 1992). More detailed analyses will be conducted and reported in a separated paper, as described in the original manuscript. In revised version of our manuscript, we add possible factors for the AMOC change due to OVL forcing to the main text.

Page 8 line 32: Here change in sea surface density flux over the North Atlantic (Speer and Tziperman, 1992) is one of possible controlling factors for the OVL-induced AMOC change. In response to the prescribed boundary conditions, changes in air and surface water temperature, atmospheric humidity, cloud cover, and surface wind speed can influence on sea surface heat fluxes (sensible heat flux, latent heat flux, and longwave and shortwave radiative flux). In addition, changes in river runoff, sea ice melt, and precipitation minus evaporation can affect sea surface salinity. These heat and salinity fluxes possibly modulate AMOC strength in the Pliocene climate. We plan to address physical processes contributing to the OVL-induced stronger AMOC in a separated paper.

*3. Page 8, line 29: how do the ice sheet and OVL enhance northward heat transport?*

The anomalous atmospheric northward heat transports in Ice Sheet effect and OVL effect are found over 50N-70N. Here these anomalous transports are not attributed to zonal-mean meridional circulation because these changes are quite small (Fig. 6d). Here the mid-latitude eddy transport may play an important role in this anomalous heat transport. Ulbrich et al. (2009)

summarized physical processes responsible for change in mid-latitude eddy activity in projected future climate. They revealed that changes in meridional temperature gradient in the middle-upper troposphere and near the surface can result in changes in jet stream and the mid-latitude eddies. Li et al. (2015) showed substantial changes in meridional temperature gradient in the troposphere in selected PlioMIP1 models. Actually, change in tropospheric meridional temperature gradient in our simulation is similar to Li et al. (2015). Such changes in meridional temperature gradient (Fig. 6a) can be one of factors for the anomalous meridional heat transport due to the atmosphere. In addition, anomalous orography over Greenland as a part of Ice Sheet effect may also contribute to changes in jet stream and mid-latitude heat transport. We add discussion on the physical processes associated with the enhanced heat transport.

Page 9 line 20: Over 50° N–70° N, OVL and Ice Sheet enhance northward heat transport, contributing to the Arctic warming (70° N–80° N). Here changes in MMC over 50° N–70° N are limited in these experiments (Fig. 6d), implying an important role of mid-latitude eddies. Changes in meridional temperature gradient in the upper troposphere and near the surface (e.g. Li et al., 2015) are possible factors for the anomalous mid-latitude eddy activity (e.g. Ulbrich et al., 2009). OVL effect contributes to an enhanced (a reduced) meridional temperature gradient in the upper troposphere (near the surface; Fig. 6a), similar to results of PlioMIP1 AOGCMs (Figs. 6 and 7 in Li et al., 2015). Such temperature changes imply a possible intensification of mid-latitude eddy activity (e.g. Mizuta, 2012). In addition, orography changes as parts of OVL and Ice Sheet effects (Fig. 3) can also affect mid-latitude atmospheric circulation and associated meridional heat transport.

*4. The OVL causes significant change in the tropical precipitation. Any idea about which factor contributes the most, orography, vegetation or lake?*

In our opinion, vegetation effect is substantially important for the tropical precipitation anomaly. Willeit et al. (2013) and Zhang and Jiang (2014) showed that change in vegetation cover from pre-industrial to Pliocene condition results in significant change in tropical precipitation. Albedo reduction over the Northern Hemisphere high latitude due to land cover change (e.g. northward shift of boreal forest; Figs. 1, 6e, 7c) contributes to the Arctic warming amplification (Figs. 5c, 6a). Associated feedback processes (e.g. snow and sea ice) amplify the high latitude warming, leading to the substantial change in meridional warming gradient (discussed in Sect. 4). Resultant inter-hemispheric warming asymmetry (larger warming over the Northern Hemisphere than the Southern Hemisphere; found in OVL effect; Fig. 6a) can lead to change in

strength and position of the intertropical convergence zone (ITCZ)-related precipitation including the increase in tropical North Atlantic-North African precipitation (Fig. 8c; e.g. Zhang and Delworth 2006; Braconnot et al. 2007). Other processes (evaporation change over tropical and subtropical land due to changes in vegetation and lakes, and orography-induced atmospheric circulation changes) may also play roles in the tropical precipitation change. We reinforce discussion on the physical processes in the main text.

From a regional perspective, tropical precipitation changes in the Pliocene run may strongly be related to changes in individual monsoon systems. In the PlioMIP2, systematic investigations of regional monsoon behaviors during the late Pliocene are planned to conduct. Detailed physical processes (e.g. Zhang R. et al. 2013) will be examined in future studies.

Page 7 line 19: Over the Northern Hemisphere high latitude, the reduced ice sheets (Figs. 1, 3) and a reduction of surface albedo (Figs. 6e, 7c) due to northward shift of boreal forest (deciduous conifer, tundra and bare soil regions over northern Canada and northeastern Eurasia; Fig. 1) result in regional warming (Fig. 5c, d). Surface snow cover (not shown) also affects partly the land surface albedo. The high latitude SAT is more sensitive to imposed albedo change due to the altered vegetation cover (Fig. 1) than low latitude (Davin and de Noblet-Ducoudré, 2010), consistent with the substantial Arctic warming found in current study (Fig. 5) and previous studies (Willeit et al., 2013; Zhang and Jiang, 2014).

Page 7 line 32: The anomalous middle and high latitude warming associated with the altered boundary conditions (e.g. darker land surface due to northward shift of boreal forest; Fig. 1) could affect the large-scale precipitation pattern via changing atmospheric circulations (Sect. 3.3). For example, tropical precipitation associated with Intertropical Convergence Zones (ITCZs) is sensitive to inter-hemispheric warming asymmetry (e.g. Braconnot et al., 2007). In the tropical Atlantic, inter-hemispheric warming gradient (warmer in the North Atlantic than the South Atlantic; Fig. 5g) affects the Atlantic and African ITCZ precipitation (Fig. 8c; e.g. Zhang and Delworth, 2006).

Page 8 line 8: The tropical precipitation changes may also be associated with changes in regional land–sea temperature contrast and seasonal monsoon circulations (e.g. R. Zhang et al., 2013).

*5. In the ice sheet experiment, how is the ice sheet defined? By changes in albedo and*

*topography? In the lake experiment, how is the lake defined in the model? These should be explicitly explained in the paper.*

*6. The effect of ice sheets should include the effect of its topography. In the OVL experiment, the effect of the ice sheet topography seems to be also included. In this case, there should be an overlap of the effect of ice sheet topography in these two experiments. Is it true?*

In this study, we estimated the ice sheet effect by comparing results of simulations with and without the ice sheets over the part of Greenland and Antarctic Continent (Figs. 1, 3). Here the anomalous ice sheets were prescribed as anomalous orography and land cover, so the resultant ice sheet effect contains both of the orography and land cover effects. The derived ice sheet effect is not overlapped with OVL effect because "orography effect" in OVL does not include the anomaly in ice sheet orography.

In our original manuscript, details of the prescribed boundary conditions were not sufficiently explained. We revise the explanations on the experimental setups.

> Page 4 line 21: We prescribed the Pliocene ice sheets by changing land orography (Fig. 1) and land cover (Fig. 3) over the ice sheets regions.

> Page 5 line 2: pre-industrial run but with Pliocene ice sheets (Ei280 run); and pre-industrial run but with Pliocene orography, vegetation, and lakes (OVL) except over the ice sheets regions (hereafter Eo280 run)

The lakes are treated in land surface model as inland water in grid boxes with a drainage basin unconnected to oceans (Yukimoto et al. 2006). The subgrid lake parameterization included in land surface model predicts water budget, but the lake surface temperature is predicted by the heat budget at the water surface, assuming a slab with a thickness of 50m, as described in Sect. 2.1. In the control run, five lakes were modeled as inland waters (Fig. 2a; Yukimoto et al. 2006). In the Pliocene run, additional lake areas are implemented in land surface (Fig. 2b). We add explanations of the treatment of lakes to Sect. 2.1.

> Page 3 line 21: Lakes are treated in the land surface model as inland water in grid boxes with a drainage basin unconnected to oceans.

> Page 4 line 29: We prescribe the Pliocene lakes (Fig. 2b; Table 2) by adding anomalous areas of lakes (Fig. 2c) to model's modern lakes (the Caspian Sea, the Aral Sea, Lake Balkhash, Lake Chad and Lake Eyre; Fig. 2a).

*7. Page 9, line 19: Please explain what the "nonlinear residual" means in terms of physics.*

The nonlinear residual term defined as Eq. 6 in this study means a departure from 'linear additivity of forcing-response relationships' (e.g. Shiogama et al. 2013). If external forcing A and B were added to system separately or together, climate response to "A+B" forcing is not necessarily identical to sum of climate responses in A forcing run and B forcing run. In case for discussion in Page 9 line 19, sea ice reduction and surface warming are larger in Sum than All (Fig. 6f). Over the regions with the Arctic sea ice edge, climatological sea ice concentration is limited. Sum of sea ice reductions due to strong forcings A and B (for example, 12% and 8%) can be larger than climatological sea ice concentration (for example, 15%), resulting in nonlinear forcing-sea ice reduction relationship (-20% is larger than -15%) because sea ice concentration is always larger than or equal to 0%. We add explanation on possible physical processes responsible for the nonlinear residual term discussed here.

> Page 10 line 9: Note that Sum overestimates the sea ice reduction and surface warming over the Northern Hemisphere high latitude simulated in All (Fig. 6a, b, f). In response to the strong external forcing including $CO_2$ and OVL, sea ice concentration can be 0 % at the edge of sea ice cover in the control climate (Howell et al., 2015). The limited sea ice concentration in the pre-industrial run is one of the possible reasons for the nonlinear relationship between forcing and sea ice reduction.

*8. Page 5, line 10: would there be any difference between with and without the addition of deep ocean temperature to the initial condition?*

The anomalous deep ocean temperature imposed in PlioMIP1 run may play some roles, but general ocean warming pattern and AMOC change are generally similar between PlioMIP2 and PlioMIP1 (Figs. 5e, 6h in this paper, and Fig. 8e in Kamae and Ueda 2012), suggesting that the imposed deep ocean temperature as initial condition is not the dominant contributor for the simulated Pliocene climate anomaly, at least after the long-term integrations (500 years). We add a note on influence of the imposed deep ocean temperature as initial condition.

> Page 5 line 17: Despite the difference in the initial deep ocean temperature, ocean circulation and SST after the 500-yr integrations are generally similar between the two (see Sects. 3.3, 4, 5).

*9. Page 5, line 26: what does the "nonlinear" mean exactly?*

The "nonlinear" effect stated here means nonlinear forcing-response relationship. Please see response to comment #7. We add meaning of "nonlinear" effect to Sect. 2.3.

Page 6 line 4: Residual (Eq. 6) indicates a nonlinear effect of combination of the boundary conditions associated with nonlinearity in forcing-response relationship (e.g. Shiogama et al., 2013).

*10. Page 5, line 29: does it suggest no interactive effect of the CO2, OVL and ice sheets?*

The interactive effect suggested by Residual term is generally minor to All, but it is not negligible for regional climate anomalies. We a note to Sect. 6 to clarify that Residual can be found in regional scale.

Page 11 line 15: However, linear additivity does not hold so well for regional climate responses including sea ice reduction over the high latitude oceans.

*11. Page 8, line 12: isn't better to change "suggesting" to "resulting from"?*

We revise as "resulting from" according to your comment.

*12. Fig11: how was the confidence level (medium, high, very high) defined?*

The confidence level shown here was derived from Dowsett et al. (2013). In their paper, confidence level of proxy-suggested climate anomaly was defined by chronology, sampling density, sampling quality and performance of quantitative method. Details of confidence scheme used in the PRISM SST reconstruction can be found in Dowsett et al. (2013). We add a note on definition of confidence level to the caption.

Figure 11 caption: Sizes of plots indicate confidence levels of SST estimate based on chronology, sampling density, sampling quality and performance of quantitative method (Dowsett et al., 2013).

*13. Page 10: lines 2-3: PlioMP2 is cooler than PlioMP1 over N. Atlantic, so the underestimation of the warming there is actually increased but not reduced as written by the*

*authors. And why A.Atlantic is cooler in PlioMP2 than PlioMP1?*

We guess that this is a reviewer's misunderstanding on the meaning of Fig. 11. SST in PlioMIP2 run is warmer than PlioMIP1 run (Figs. 6b, 11a), so the underestimation of the warming is reduced (Fig. 11b). Generally PlioMIP2 run shows larger warming than PlioMIP1 run (Table 3). Both the underestimations of warming over the North and South Atlantic middle and high latitude (Fig. 11a) are reduced in the PlioMIP2 run (Fig. 11b). We revise main text in section 5 to avoid confusing.

> Page 10 line 27: Figures 11 and 12 compare SST biases found in the PlioMIP2 and 1. Generally, both the PlioMIP2 and 1 tend to underestimate the mid-to-high-latitude warming suggested by proxy records (blue circles in Fig. 11a; Haywood et al., 2013). However, the large part of underestimation of the mid-to-high-latitude warming is reduced substantially in the PlioMIP2 run (green circles in Fig. 11b).

*14. Page 10, line 15: the linear additivity of the Pliocene climate simulation is not necessarily obvious at regional scale (see fig6).*

We agree to your comment. Global-mean response shows good linear additivity (Table 3), but the linear additivity is limited for the regional climate responses. We add a note to this section.

> Page 11 line 15: However, linear additivity does not hold so well for regional climate responses including sea ice reduction over the high latitude oceans.

*15. Page 10, line 21: Please comment what are the possible reasons that the model fails to reproduce the extremely warm condition over the Arctic to high-latitude North Atlantic region.*

This underestimation of the high latitude North Atlantic warming was also found in PlioMIP1 multiple climate models. Haywood et al. (2013) pointed out that almost all the PlioMIP1 AOGCMs underestimated this extreme warming. Possible reasons why the model cannot reproduce the extremely warm condition over the high latitude North Atlantic may be associated with (1) AMOC biases in models, (2) sea ice biases in models, and/or (3) uncertainty in SST estimate based on proxy records (Robinson 2009; Haywood et al. 2013). Haywood et al. (2013) noted that this large model/data discord was highly dependent on geochemically-based proxy mean annual temperature estimate and was not derived from faunal based estimates of cold/warm month means. They suggested that we should not rely on this model/data discord too

much until more variety of proxy records is available from more locations in the high latitude North Atlantic and the Arctic. We add discussion on this issue to section 5.

[revised manuscript text omitted]